# The SpxA1-TenA toxin-antitoxin system regulates epigenetic variations of *Streptococcus pneumoniae* by targeting protein synthesis

**Shaomeng Wang**[1�], **Xiu-Yuan Li**[1�], **Mengran Zhu**[1], **Haiteng Deng**[2], **Juanjuan Wang**[3]*, **Jing-Ren Zhang**[1]*

**1** Center for Infection Biology, School of Basic Medical Sciences, Tsinghua University, Beijing, China, **2** MOE Key Laboratory of Bioinformatics, School of Life Sciences, Tsinghua University, Beijing, China, **3** College of Veterinary Medicine, China Agricultural University, Beijing, China

360 These authors contributed equally to this work.
* juanjuanwang@cau.edu.cn; zhanglab@tsinghua.edu.cn

**Data Availability Statement:** All the data involved in this study are available in the manuscript, supplementary figures and tables. The raw data of SMRT sequencing and RNA sequencing are

## Abstract

Human pathogen *Streptococcus pneumoniae* forms multiple epigenetically and phenotypically distinct intra-populations by invertase PsrA-driven inversions of DNA methyltransferase *hsdS* genes in the colony opacity-determinant (*cod*) locus. As manifested by phase switch between opaque and transparent colonies, different genome methylation patterns or epigenomes confer pathogenesis-associated traits, but it is unknown how the pathogen controls the *hsdS* inversion orientations. Here, we report our finding of the SpxA1-TenA toxin-antitoxin (TA) system that regulates the orientations of *hsdS* inversions, and thereby bacterial epigenome and associated traits (e.g., colony opacity) by targeting pneumococcal protein synthesis. SpxA1 and TenA were found to constitute a highly conserved type II TA system in *S. pneumoniae*, primarily based on the observation that overexpressing toxin TenA led to growth arrest in *E. coli* and enhanced autolysis in *S. pneumoniae*, and the antitoxin SpxA1 repressed the transcription of the *spxA1-tenA* operon. When the transcription of *tenA* was de-repressed by a spontaneous AT di-nucleotide insertion/deletion in the promoter region of the *spxA1-tenA* operon, TenA bound to the ribosome maturation factor RimM, and thereby reduced the cellular level of alternative sigma factor ComX (known for the activation of natural transformation-associated genes). Attenuation of ComX expression in turn enhanced the transcription of the invertase gene *psrA*, which favored the formation of the transparent colony phase-associated *hsdS* allelic configurations in the *cod* locus. Phenotypically, moderate expression of TenA dramatically reshaped pneumococcal epigenome and colony opacity. Because spontaneous variations frequently occur during bacterial growth in the number of the AT di-nucleotides in the promoter region of the *spxA1-tenA* operon, this locus acts as a programmed genetic switch that generates pneumococcal subpopulations with epigenetic and phenotypic diversity.

available at the NCBI database are available under the PRJNA1136146 and PRJNA1136171 project in the SRA database.

**Funding:** This work was supported by grants from National Natural Science Foundation of China to J-. R.Z (No.82330071) and J.W (No.32100141), and from Tsinghua Initiative Scientific Research Program to J-.R.Z (No.20243080033; 20233080054). The funders had no role in study design, data collection and analysis, decision to publish, or preparation of the manuscript.

## Author summary

*Streptococcus pneumoniae* is not only a common commensal in the upper airway of humans, but also a major cause of community-acquired pneumonia, sepsis, meningitis and otitis media, even after antibiotics and capsular polysaccharide-based vaccines have been applied for decades. It is largely undefined how *S. pneumoniae* achieves the extreme adaptability during nasopharyngeal colonization and disseminating infections, with a relatively small genome and few extrachromosomal elements. Here we show a newly identified type II toxin-antitoxin (TA) system in *S. pneumoniae* that acts as a programmed genetic switch to generate intra-populations with remarkable epigenetic and phenotypic diversity. While toxin TenA morphs *S. pneumoniae* into the transparent colony-associated genome methylation patterns and colony phase by modulating the orientations of methyltransferase *hsdS* gene inversions, antitoxin SpxA1 tilts the bacterium to the opposite physiological conditions by repressing the transcription of *tenA* and thereby countering the activity of TenA. Moreover, we have found that the functional balance between TenA and SpxA1 is controlled by the programmed AT di-nucleotide insertion/deletion in the promoter of the *spxA1-tenA* operon. This work has thus revealed a previously uncharacterized toxin-antitoxin system that generates phenotypic diversity by a combination of genetic and epigenetic mechanisms.

## Introduction

*Streptococcus pneumoniae* (the pneumococcus) is a commensal in the human nasopharynx and also an opportunistic pathogen of invasive infections (pneumonia, bacteremia and meningitis) [1]. To adapt to diverse host niches, the bacterium employs complex genetic and epigenetic mechanisms to generate phenotypically heterogeneous subpopulations. In particular, *S. pneumoniae* is capable of reversible phase variation between the opaque (O) and transparent (T) colony variants within clonal populations [2,3]. Both phases possess distinct molecular and pathogenic characteristics. The O variants express a thicker capsule with stronger resistance to host phagocytic clearance. In contrast, the T counterparts produce a thinner capsule displaying higher airway adherence [3–6]. These molecular and structural differences are associated with pneumococcal behaviors in animal models. While the O variants are more prevalent in bloodstream infections, the T counterparts are more dominant in nasopharyngeal colonization [3,4].

Pneumococcal phase variation in colony opacity is epigenetically determined by reversible DNA inversions in the three DNA methyltransferase *hsdS* genes in the Spn556II/SpnD39III type I restriction-modification (R-M) system or the **c**olony **o**pacity **d**eterminant (*cod*) locus [7–9]. The *cod* locus contains the *hsdR* (restriction endonuclease), *hsdM* (DNA methyltransferase), *psrA* or *creX* (DNA invertase), $hsdS_A$ (sequence recognition), and two transcriptionally silent *hsdS* genes ($hsdS_B$ and $hsdS_C$) [7–9]. As a tyrosine recombinase, PsrA catalyzes extensive DNA inversions between $hsdS_A$ and the two silent homologs ($hsdS_B$ and $hsdS_C$) by recognizing the inverted repeats (IRs) flanking the invertible sequences. The *hsdS* inversions generate six bacterial subpopulations, each of which carries a different $hsdS_A$ allele ($hsdS_{A1}$ to $hsdS_{A6}$) [7,9,10]. Each of the six $HsdS_A$ subunits drives the N6-methyladenine (6-mA) methylation of a unique sequence motif that is present in many genomic loci, forming a distinct methylome [7,9]. Phenotypically, only the pneumococci carrying $hsdS_{A1}$ allele produce O colonies, while the other five $hsdS_{A2-6}$-carrying variants generate T colonies [7,9]. Each of the six $hsdS_A$ alleles defines a unique transcriptome [7]. Our recent study has shown that the orientations of *hsdS*

inversions in the *cod* locus and resulting colony phases are subjective to transcriptional regulation by four two-component regulatory systems (TCSs) [11]. The absence of these TCSs leads to dramatic loss of $hsdS_{A1}$-carrying variant and thereby O colonies in the clonal populations. Together, the epigenetic switches driven by *hsdS* inversions in the *cod* locus confer *S. pneumoniae* variable biological characteristics and pathogenetic identities to enhance bacterial adaptation.

*S. pneumoniae* is capable of natural genetic transformation or the competence state to achieve genetic diversity by uptaking and recombining foreign DNA molecules into the genome. The competence state is transiently activated by the ComDE, a two-component system in response to an extracellular competence stimulating peptide (CSP) [12]. Exposure to CSP activates the transcription of *comX*, *comW* and other early competence genes that are responsible for further activation of the late competence genes for DNA uptake and processing [13,14]. ComX is the alternative sigma factor that, together with ComW, activates the late competence genes for DNA uptake and processing as a part of natural transformation by binding to the "*com* box" sequence in the promoter regions of the target genes [13,15,16]. There are two identical copies of *comX* (*comX1* and *comX2*) at two remote loci of pneumococcal genome. The cellular level of ComX is stabilized by ComW against otherwise rapid degradation by the ClpP protease [17]. Our previous study has shown that ComW, but not the two-component system ComDE, is required for stabilizing the $hsdS_{A1}$ allelic configuration in the *cod* locus by an uncharacterized mechanism [11].

The toxin-antitoxin (TA) systems consist of stable toxins and labile antitoxins, and widely exist in bacteria [18]. Toxins inhibit bacterial growth by targeting essential cellular processes, whereas antitoxins neutralize the activities of cognate toxins at the levels of transcription, translation, activity or stability under steady-state conditions. Under the conditions when toxins are outnumbered by antitoxins, the toxic activity of the toxins is unleashed. Based on the nature of the antitoxin and the mode of interaction between the toxin and antitoxin, TA systems are currently classified into eight types, in which the type II TA systems are the most prevalent [19]. A typical type II TA system consists of toxins and antitoxins both in protein form, whose encoding genes are organized as an operon. In the most cases, type II toxins target protein synthesis by degrading mRNAs or tRNAs, impairing aminoacylation of tRNA and inactivating elongation factor [20–22]. While up to 10 type II TA systems are bioinformatically predicted in *S. pneumoniae* [23], only a few TA systems are characterized [19]. RelBE, YefMB and HicBA appear to target pneumococcal translation, whereas the PezT toxin of the PezAT system inhibits cell wall synthesis [19,24].

In this study, we have serendipitously discovered a new type II toxin-antitoxin system in *S. pneumoniae*, SpxA1-TenA system, which was not included in the previously predicted type II TA systems [19]. The *tenA* (toxin) and *spxA1* (antitoxin) genes are co-transcribed as an operon. The balance of the SpxA1-TenA system is controlled at the transcriptional level by the variable length of the AT dinucleotide repeat (AT-rich) region upstream of the *spxA1-tenA* promoter. Excessive TenA profoundly affects pneumococcal biology in DNA methyltransferase *hsdS* gene configuration, genome methylation pattern and the resultant colony phase.

## Results

### TenA promotes pneumococcal transparent colony phase

Our previous study has showed that two-component system TCS06, consisting of the sensing kinase HK06 and response regulator RR06, regulates pneumococcal methylome and the resulting phase switch between opaque (O) and transparent (T) colonies within clonal populations [11]. During that work, an *rr06* deletion mutant Δ*rr06* (TH9164) produced significantly more

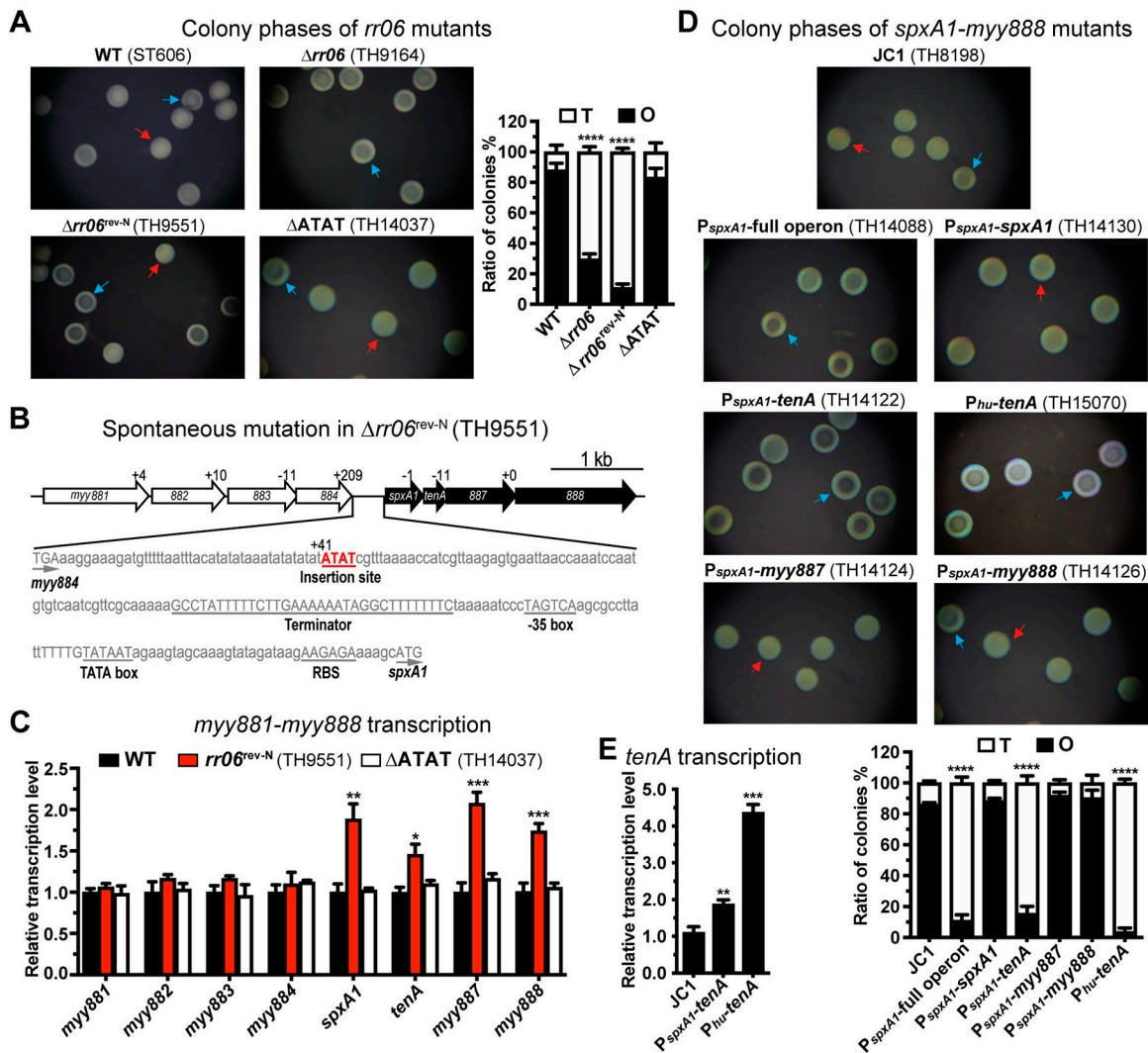

**Fig 1. Functional impact of TenA on colony opacity of *S. pneumoniae*.** (**A**) Colony phenotypes and ratios between opaque (O) and transparent (T) colonies of WT (ST606) and isogenic *rr06* mutants. Representative colonies are indicated by red (O) and blue (T) arrowheads, respectively. Genotype (top) of each strain is marked. The mean value ± s.d. of three plates for the O and T colony ratio is presented in a single bar. (**B**) Schematic diagram of the insertion mutation in Δ*rr06*^rev-N. The nucleotides between two adjacent genes are marked in base pairs. (**C**) The relative mRNA abundance of *myy881-888* in WT, *rr06*^rev-N and ΔATAT mutants was detected by qRT-PCR with *era* gene as an internal control, and shown as mean value ± s.d. of three replicates. The mRNA levels were presented as relative values to that of WT (ST606). (**D**) Colony phenotypes and ratio between O and T colonies of the *spxA1* operon overexpressed variants. (**E**) Relative abundance of *tenA* mRNAs in *tenA*-overexpressed variants was detected and presented as in (C).

T than O colonies, but the phenotype could not be genetically complemented with an intact *rr06* in the complementation strain Δ*rr06*^rev-N (TH9551) (**Fig 1A**). The *in situ* replenishment did not change the T colony dominance. This indicated that other gene(s) beyond *rr06* were involved in colony formation. Genome sequencing identified a four-nucleotide 5'-ATAT-3' insertion between the 41st and 42nd nucleotides after the stop codon of *myy884* in strain Δ*rr06*^rev-N (**Fig 1B**). We thus constructed an unmarked deletion mutant of the 5'-ATAT-3' insertion in parental strain Δ*rr06*^rev-N. The deletion strain ΔATAT (TH14037) showed a similar O colony-dominant phenotype as WT strain, which produced 83.3% of O colonies and 16.7% of T colonies (**Fig 1A**). These results suggested that the 5'-ATAT-3' insertion modulates the colony phase switching.

Subsequent sequence analysis revealed that the inserted sequence is located in the 209-base pair (bp) intergenic region between two gene clusters *myy881-884* and *spxA1-myy888*. The upstream genes encode four hypothetical proteins, while the downstream cluster encodes SpxA1 and three uncharacterized proteins. SpxA1 has been reported to repress the expression of pneumococcal competence genes [25]. Sequence analysis predicted a Rho-independent transcriptional terminator after the stop codon of *myy884* and the -35 to -10 promoter motifs upstream of *spxA1* (**Fig 1B**). This sequence configuration suggested that the inserted sequence may impact colony phenotype by altering the expression of the adjacent genes. We thus tested this hypothesis by comparing the expression of the two operons between strains Δ*rr06*^rev-N and ΔATAT. The result revealed a substantial increase in the transcription of *spxA1*, *myy2735* (*tenA*), *myy887* and *myy888* in strain Δ*rr06*^rev-N, which was restored to the WT levels in ΔATAT variant after the 5'-ATAT-3' deletion (**Fig 1C**). As a control, the transcripts of each gene in cluster *myy881-884* did not show obvious change. Such a simultaneous increase in the mRNA level among the *spxA1-myy888* gene cluster indicated that these four genes are co-transcribed as an operon by the common promoter upstream of *spxA1*, which is consistent with the proximity among these genes (**Fig 1B**).

To figure out which gene(s) are responsible for the alteration of colony opacity, we *in trans* duplicated the entire operon or each constituent gene in the *bgaA* locus of ST556 derivative JC1 control variant (TH8198), in which *bgaA* gene was replaced by counterselection construct JC1 as described [26]. Each target sequence was placed after the entire 5' non-coding region of *spxA1*. The strains with duplication of the whole gene cluster or *myy2735* (*tenA*) alone predominantly produced T colonies (**Fig 1D**). However, the isogenic counterparts of *spxA1*, *myy887* and *myy888* still produced a comparably high level of O colonies as WT. These results showed that MYY2735 drives the T colony phenotype. Thus, we designated *myy2735* as **t**ransparent **en**hancer **A**, *tenA*. We further confirmed the result by placing *tenA* after the promoter sequence of the *hu* gene in the *bgaA* locus, which is one of the strongest promoters in *S. pneumoniae* [27]. The *hu* promoter-driven *tenA* dramatically reduced O colonies (**Fig 1D**), which agreed with a 4.4-fold increase in *tenA* transcription (**Fig 1E**). We further verified the impact of TenA on colony phase by constructing the *tenA*-overexpressed variants in the *bgaA* locus of strains P384 (serotype 6A) and ST877 (serotype 35B) [9,11]. The overexpression of *tenA* driven by *spxA1* or *hu* promoter led to significantly increased abundance of *tenA* mRNA in P384 and ST877 backgrounds (**S1A Fig**), and T colony phases (**S1B Fig**). These lines of evidence revealed that TenA promotes T colony phase.

## TenA locks the *cod* locus in the T phase-defining genetic configurations

Our previous studies have demonstrated that PsrA-catalyzed inversions of *hsdS* genes in the *cod* locus and the resultant genome methylation patterns or methylomes epigenetically determine pneumococcal colony phases (**Fig 2A**) [9–11]. To define whether TenA regulates colony phase via the *hsdS* inversions, we tested the impact of *tenA* overexpression on colony phase in the *psrA*^Y247A mutant, which carried a point mutation in the 247^th catalytic residue tyrosine and thus locked in O colony phase [11]. The overexpression of *tenA* in the *psrA*^Y247A background did not show any obvious impact on the colony phenotype (**Fig 2B**). This indicates that TenA regulates colony phase via modulating PsrA-driven *hsdS* inversions in the *cod* locus.

To characterize the impact of TenA on the *hsdS* gene configuration, we determined the fraction of the $hsdS_{A1}$ allele-carrying subpopulation in the clonal population by measuring the relative abundance of $hsdS_{A1}$ mRNAs as previously described [11]. $hsdS_{A1}$ defines O colony phase, while the other five allelic variants of the $hsdS_A$ gene in the *cod* locus promote T colony formation [9]. Compared to the positive control of the *psrA*^Y247A strain expressing only

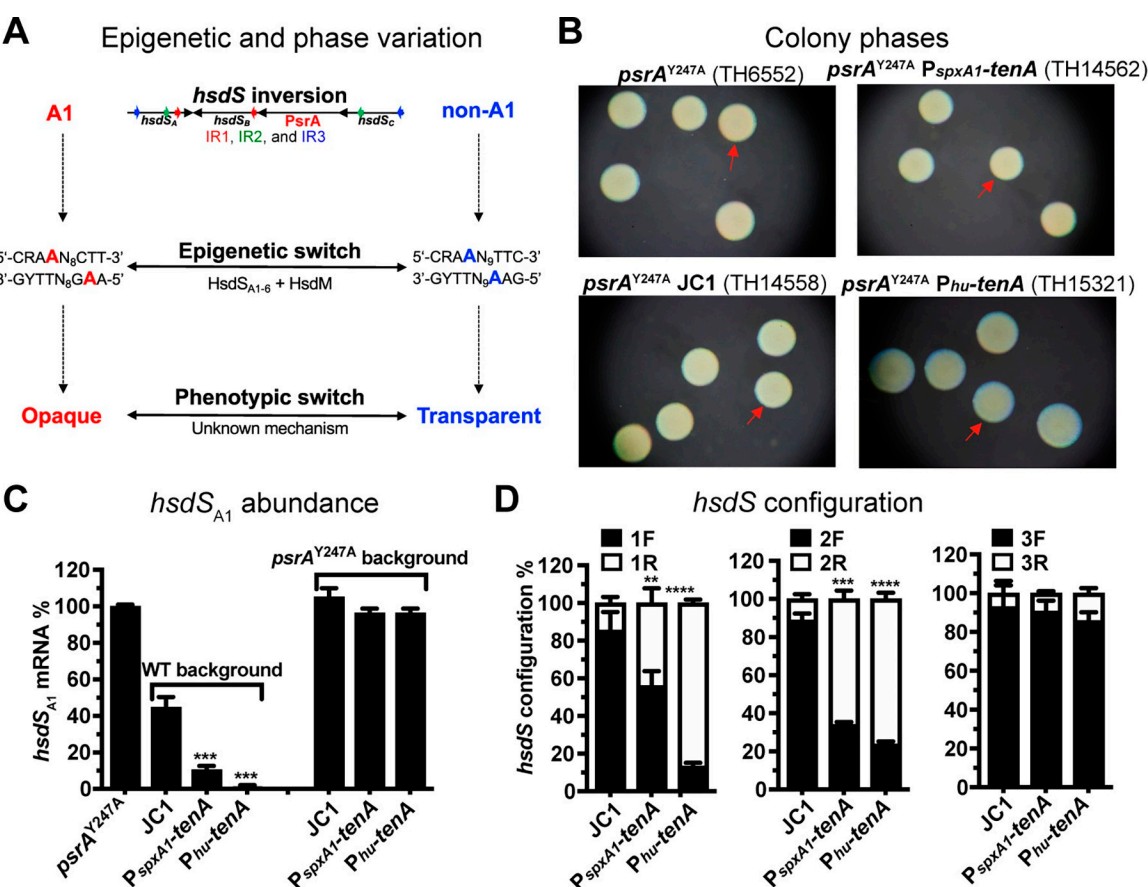

**Fig 2. Dependence of TenA on PsrA-catalyzed *hsdS* inversions.** (**A**) Diagram of *hsdS* inverton-driven phase variation in colony opacity. IR1, IR2 and IR3 are marked as red, green or blue arrowhead, respectively. The methylated adenine nucleotides in DNA motif are highlighted in red or blue. R = A or G, Y = T or C. (**B**) Colony phenotypes of the *tenA*-overexpressing variants in *psrA*[Y247A] background. Red arrowheads indicate the opaque (O) colonies. (**C**) Relative abundance of the *hsdS*[A1] mRNAs in *tenA*-overexpressed variants in WT and *psrA*[Y247A] background. The mRNA levels of the *hsdS*[A1] allele relative to that of the non-invertible *hsdS* in the clonal populations of each strain were detected by qRT-PCR and normalized to the value of *psrA*[Y247A] strain. Data are shown as the mean value ± s.d. of three replicates. (**D**) The *hsdS* allelic configurations in the *tenA*-overexpressed variants. The ratios of forward (F) and reverse (R) orientations of each IR-bounded sequence were measured by qPCR. The ratio of bacteria with IRs in different orientations in each mutant is shown as mean value ± s.d. of three replicates in a representative experiment.

*hsdS*[A1], the JC1 control strain showed 44.5% *hsdS*[A1]-carrying bacteria. Consistent with the significant reduction in O colonies, the *hsdS*[A1] transcripts were dramatically diminished in isogenic strains carrying P[spxA1]- or P[hu]-driven *tenA* (**Fig 2C**). By contrast, the overexpression of *tenA* in the *psrA*[Y247A] background did not alter the *hsdS*[A1] abundance, which produced a comparable level of *hsdS*[A1] mRNAs as the parental *psrA*[Y247A] strain (**Fig 2C**). This indicates that TenA promotes the T phase (non-*hsdS*[A1]) configurations in the *cod* locus.

To elaborate how TenA modulates the *hsdS* allelic configurations in the *cod* locus, we quantified the DNA inversion frequency mediated by each inverted repeat (IR) through measuring the fraction of either forward (F) or reverse (R) orientation with diagnostic primers as described in the previous study (**S2 Fig**) [10]. Consistent with the abundant *hsdS*[A1] mRNAs and O-dominant clonal phenotype, the JC1 control strain showed an obvious opaque allelic *hsdS*[A1] configuration, in which the majority of the IR1-, IR2- and IR3-bound sequences displayed forward orientations (**Fig 2D**). By comparison, the overexpression of *tenA* significantly decreased the forward fractions of both IR1- and IR2-bound sequences, indicative of the

opaque configuration. The proportions of IR1- and IR2-forward orientations reduced from 85.5% and 88.8% in parental strain to 56.0% and 34.0% in $P_{spxA1}$-*tenA* variant, respectively (**Fig 2D**). These gene configurations were further reduced in the strain that expressed *tenA* under the control of the *hu* promoter. Only 13.3% and 23.8% of IR1- and IR2-forward populations were detected, respectively (**Fig 2D**). Taken together, these data demonstrates that TenA promotes the formation of T colony phase by locking the *cod* locus in a T phase-defining genetic configuration.

## TenA turns off the O phase-defining epigenome

Because only the $hsdS_{A1}$ allele encoding HsdS$_{A1}$ MTase defines the O phase methylome [9,28], the abovementioned changes of the *tenA*-overexpressed variants in $hsdS_A$ allele genotypes and colony phenotypes strongly suggested that TenA modulates pneumococcal methylome. We thus performed genome-wide detection of N6-methyladenine (6-mA) in the *tenA*-overexpressed strain by single molecule real-time (SMRT) sequencing (**Fig 3A**) [8]. This trial detected 6-mA methylation for virtually all copies of the Spn556I (5'-TCTAG$^{m6}$A-3', Type II R-M) and Spn556III (5'- GAT$^{m6}$AN$_7$TCA-3', Type I R-M) recognition motifs in both the WT and the isogenic *tenA*-overexpressed variant (**S2** and **S3 Tables**). However, there were dramatic differences in the methylome of the *cod* locus (Spn556II) between these two strains.

Consistent with the previous findings that three of the six $hsdS_A$ allelic variants ($hsdS_{A1}$, $hsdS_{A2}$ and $hsdS_{A3}$) are dominantly present [7,9,11], SMRT sequencing revealed 6-mA

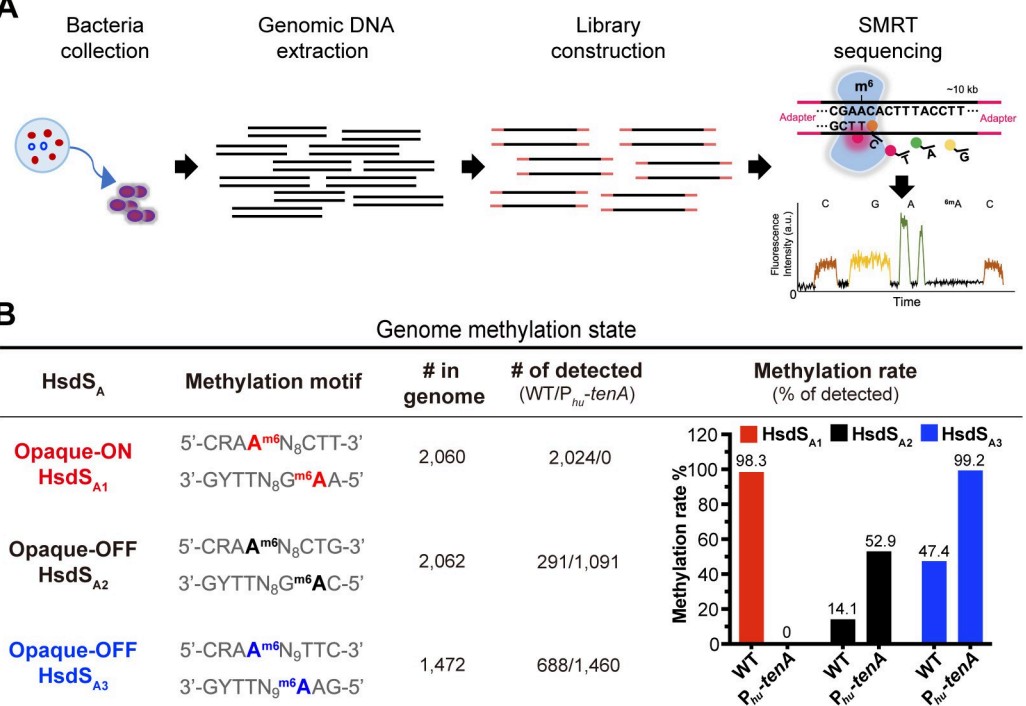

**Fig 3. Enrichment of T phase-defining methylomes by TenA.** (**A**) Illustration of detecting the genomic DNA methylation by SMRT sequencing. (**B**) Relative methylation rate of the DNA motifs recognized by three HsdS$_A$ allelic variants in WT and *tenA*-overexpressed variant. The methylation sequences specified by each HsdS$_A$ MTase were detected by SMRT sequencing. "# in genome" indicates the total number of loci on both DNA strands in the ST556 genome (accession CP003357.2). "# of detected" and "% of detected" represent the number and ratio of loci detected by SMRT sequencing, respectively.

methylation in the motifs recognized by only three of the six $hsdS_A$ allelic variants (HsdS$_{A1}$, HsdS$_{A2}$ and HsdS$_{A3}$) in WT strain (**Fig 3B**). Virtually all 2,060 loci of the HsdS$_{A1}$ motif in the genome of WT were methylated (98.3%), but the methylation percentages for HsdS$_{A2}$ and HsdS$_{A3}$ motifs were much lower (14.1% and 47.4%, respectively). No 6-mA methylation was detected at any loci of the HsdS$_{A4}$, HsdS$_{A5}$ and HsdS$_{A6}$ motifs. In sharp contrast, the methylated HsdS$_{A1}$ motif completely disappeared in *tenA*-overexpressed variant P$_{hu}$-*tenA*, which was consistent with its non-*hsdS$_{A1}$* gene configuration and T-dominant colony phenotype. Accordingly, the methylation rates of non-HsdS$_{A1}$ motifs significantly increased to 52.9% for HsdS$_{A2}$ and 99.2% for HsdS$_{A3}$ in P$_{hu}$-*tenA* variant compared with the WT counterpart. These SMRT sequencing data further confirmed that TenA promotes the T colony phase formation through enriching the non-HsdS$_{A1}$ methylome. Taken together, TenA triggers the PsrA-catalyzed IR1- and IR2-mediated *hsdS* inversions to non-*hsdS$_{A1}$* state and results in the T phase methylome, which finally generating a T-dominant colony phase.

## The *spxA1-tenA* operon encodes a new type II TA system

TenA is annotated as a hypothetical protein without any known biological function in the pneumococcal genomes [29–32]. Database search suggested that TenA (92 amino acids) and SpxA1 (133 residues) constitute a new type II toxin-antitoxin system. TenA and SpxA1 share 59.8% and 73.7% amino acid sequence identities with the toxin GinC (accession WP_002984497) and its antitoxin (accession WP_010922362) of *Streptococcus pyogenes*, respectively (**S3A Fig**). These result suggest that TenA and SpxA1 represent an uncharacterized TA system. The toxin and antitoxin genes in type II TA systems are typically arranged as an operon, with the first gene encoding the antitoxin [33,34]. In agreement with the overlapping feature of the *spxA1* and *tenA* coding regions (**Fig 4A**), the RT-PCR data revealed that *spxA1* and *tenA* are transcribed as an operon (**Fig 4A**). *spxA1* and *tenA* were co-amplified as a 680-bp product from the cDNA template using primers targeting the 5' end of *spxA1* (F1) and the 3' end of *tenA* (F4). Another feature of type II TA systems is their transcriptional self-regulation [33,34]. We thus determined whether SpxA1 regulates the operon by detecting the *tenA* mRNAs in the Δ*spxA1* mutant. The absence of *spxA1* led to significant increase in the level of the *tenA* mRNAs by 2.7-fold as compared with that of the parental strain. Consistently, the *tenA* transcript in Δ*spxA1* was completely restored to that of the parental strain by *in situ* restoration of *spxA1* (**Fig 4B**). To verify the transcriptional repression of SpxA1 to the *spxA1-tenA* operon, we overexpressed *spxA1* under the control of the *hu* promoter in a neutral *bgaA* locus. The RT-qPCR result demonstrated that the *tenA* transcription was significantly repressed in the *spxA1*-overexpressing strain as compared with that of the JC1 control (**Figs 4C and S4A**). These lines of evidence show that SpxA1 autoregulates the *spxA1-tenA* operon, which further supports the toxin-antitoxin nature of this locus.

*S. pneumoniae* also possesses a homolog of SpxA1—SpxA2. The two proteins share 45.9% amino acid sequence identity (**S3B Fig**). Turlan *et al.* show that SpxA1 and SpxA2 have a redundant function, as deletion of *spxA2* is synthetically lethal in the absence of *spxA1*. However, these two proteins have their unique functions in regulation of competence genes (for SpxA1) and cell growth (for SpxA2) [25]. We determined whether SpxA2 impacts *spxA1-tenA* expression and *hsdS$_{A1}$* abundance. The Δ*spxA2* mutant showed a comparable level of *tenA* mRNA as the parental strain, and a modest increase in the *spxA1* transcription (**S4B Fig**). Surprisingly, there was a significant reduction in the proportion of the opaque colony-defining *hsdS$_{A1}$* allelic variant in the Δ*spxA2* mutant (**S4C Fig**). This finding strongly suggests that SpxA2 affects the *hsdS* inversions through a mechanism that is independent of the SpxA1-TenA system.

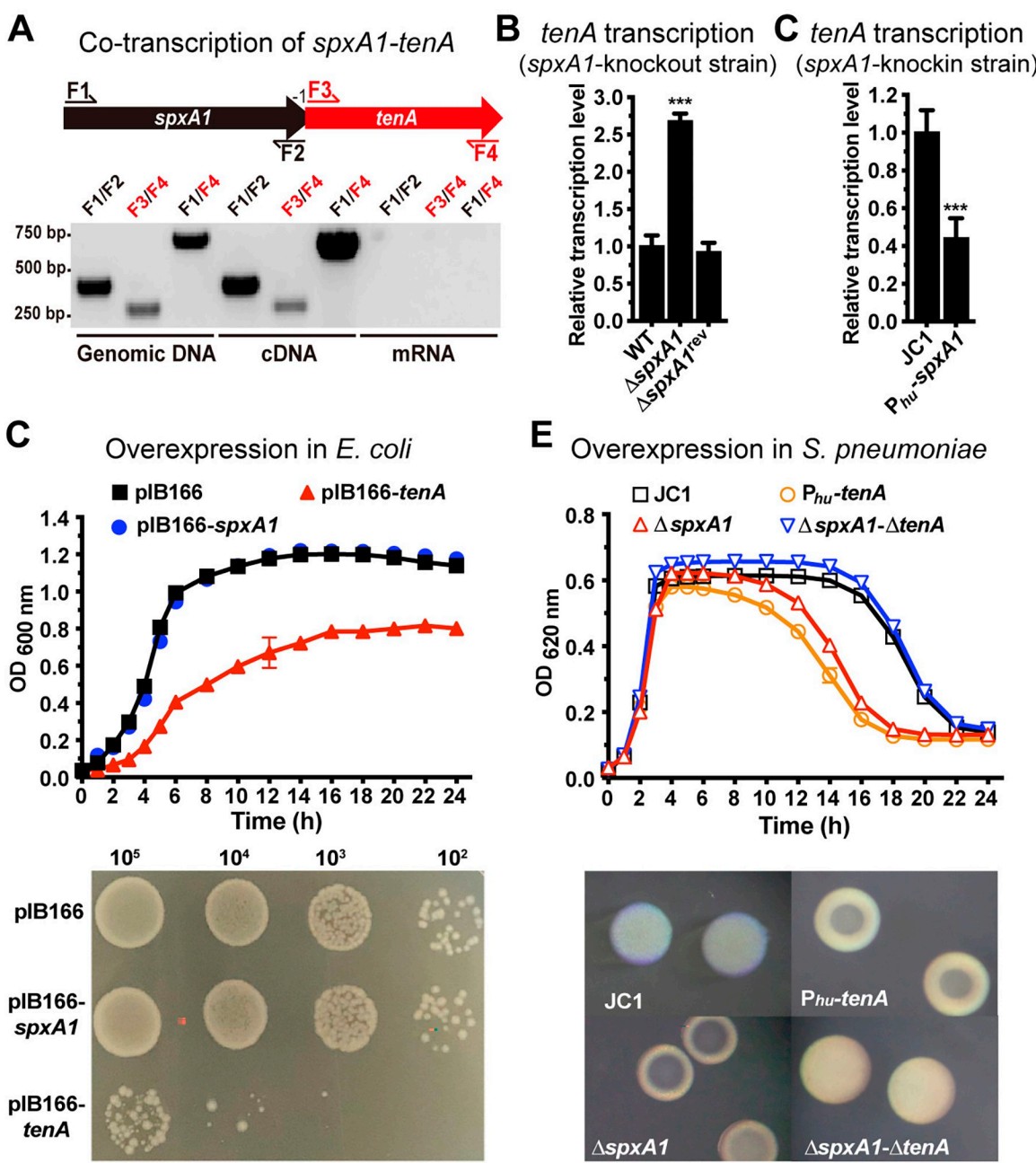

**Fig 4. Functional identification of SpxA1-TenA as a toxin-antitoxin system.** (**A**) Gene organization of *spxA1-tenA* locus. Positions of the primers used for PCR amplification are indicated. For detection, primer pairs F1/F2, F3/F4 and F1/F4 were used to amplify *spxA1*, *tenA* and *spxA1-tenA*, respectively. Amplicons from WT genomic DNA as a positive control, the fragment amplified by F1/F4 from cDNA template indicated that these two genes belong to a transcription unit. Amplicons from total mRNA of WT strain as a negative control. (**B**) Repression of *tenA* transcription by SpxA1 was tested in the *spxA1* deletion mutant and its revertant by qRT-PCR as in Fig 1C. (**C**) Repression of *tenA* transcription by SpxA1 was verified by overexpressing *spxA1* with the *hu* promoter in the *bgaA* locus and performing qRT-PCR as shown in Fig 1C. (**D**) The bacterial growth was detected with the inducible expression of SpxA1 and TenA by supplementing 1 mM IPTG to determine the toxicity of TenA in *E. coli*. Bacteria carrying plasmids pIB166-*spxA1*, pIB166-*tenA* or plasmid control were cultured in LB broth to measure the growth curve by detecting the absorbance at $OD_{600\ nm}$ (upper panel), and plotted on LB plate to measure cell viability (bottom panel). (**E**) The toxicity of TenA in *S. pneumoniae* was determined by monitoring the growth curve of *spxA1-tenA* mutants at $OD_{620\ nm}$ in C+Y medium (upper panel) and colony opacity (bottom panel).

We further tested the potential toxicity of TenA as a type II TA toxin by inducible ectopic expression of *tenA* or *spxA1* in *E. coli*. Inducible expression by IPTG treatment led to significant growth inhibition in *E. coli* strain carrying the pIB166-*tenA* plasmid but not that harboring the *spxA1* counterpart (**Fig 4D**, upper panel). Consistently, bacteria expressing *tenA* but not *spxA1* displayed dramatic impairment in colony formation (**Fig 4D**, bottom panel). These data have strongly suggested TenA as a toxin. By contrast, the toxic effect of TenA in *S. pneumoniae* is more modest. Overexpression of *tenA* in P$_{hu}$-*tenA* or deletion of *spxA1* in Δ*spxA1* led to earlier autolysis at the stationary phase, although no significant impact on bacterial growth was observed with these strains at the log phase (**Fig 4E**, upper panel). This growth defect can be restored to the parental level in Δ*spxA1*-Δ*tenA* double knockout variant. The accelerated autolysis in *tenA*-overexpressed variant was verified by the overwhelming dominance of T colonies in these bacteria (**Fig 4E**, bottom panel). Commonly, a classical type II TA system forms a stable complex that renders the toxin inactive [35]. However, further investigation revealed that SpxA1 and TenA have no obvious interaction in bacterial adenylate cyclase-based two-hybrid (BACTH) system (**S4D Fig**), although the bioinformatic analysis by Alpha-Fold3 [36] strongly suggested that SpxA1 and TenA form a strong molecular complex (**S4E Fig**). These lines of evidence have strongly suggest that TenA and SpxA1 form a previously uncharacterized type II toxin-antitoxin system, in which toxin TenA is transcriptionally repressed by its antitoxin SpxA1.

## TenA-associated ComX represses the expression of the PsrA invertase

To figure out the molecular mechanism behind the action of TenA in regulation of pneumococcal epigenetic and colony phases, we performed affinity pull-down assay with His-tagged TenA to identify potential protein(s) that acts as liaison between TenA and the *hsdS* inversions. This experiment identified 65 proteins with significant enrichment (**Figs 5A and S5** and **S4 Table**). Notably, PsrA was absent from the hits, suggesting that TenA regulates the *hsdS* inversion without direct interaction with the invertase. The majority of the hits were associated with ribosomal structure (45) or biogenesis (6). We first tested the impact of the seven non-essential proteins out of the 14 non-ribosomal protein hits. Deleting each of the seven genes led to significantly decreased *hsdS*$_{A1}$ allelic configuration in clonal population, except for *cmbR* (**Fig 5B**), indicating that these genes are associated with the *hsdS* inversions. This finding was consistent with significant reduction of opaque colonies in these mutants (**S6A Fig**).

We next focused on ComX because it was the most enriched DNA-associated protein by TenA (**Fig 5A**). ComX is an alternative sigma factor that activates the expression of pneumococcal competence-associated genes for DNA uptake and processing by interacting with the "*com* box" sequence in the promoter regions of the target genes [37,38]. There are two copies of *comX* (*comX1* and *comX2*) at two remote loci in the pneumococcal genome. Deleting either *comX1* or *comX2* led to dramatic reduction in the *hsdS*$_{A1}$-carrying subpopulation (**Fig 5B**). To verify whether ComX modulates the *hsdS* inversions, we quantified the *hsdS* allelic configurations by measuring the fractions of either forward (F) or reverse (R) orientation by qPCR as described [10]. In line with the alterations of IR1- and IR2-mediated inversion frequency in the *tenA*-overexpressed variant, the fractions of IR1- and IR2-forward orientation that represent the *hsdS*$_{A1}$ allelic configuration were significantly decreased in Δ*comX1-X2* mutant (**Fig 5C**). Consistent with the O phase genotype and non-*hsdS*$_{A1}$ allelic configuration, both the *comX* single gene deletion mutants and *comX1-X2* double knockout variant produced noticeable T phase (**Fig 5D**). Similar to the *tenA*-overexpressed strain (**Fig 3B**), PacBio sequencing revealed no methylation at any of the HsdS$_{A1}$ motif loci in the Δ*comX1-X2* genome. Instead, methylation rates for the HsdS$_{A2}$ and HsdS$_{A3}$ motif loci were greatly enhanced in the absence

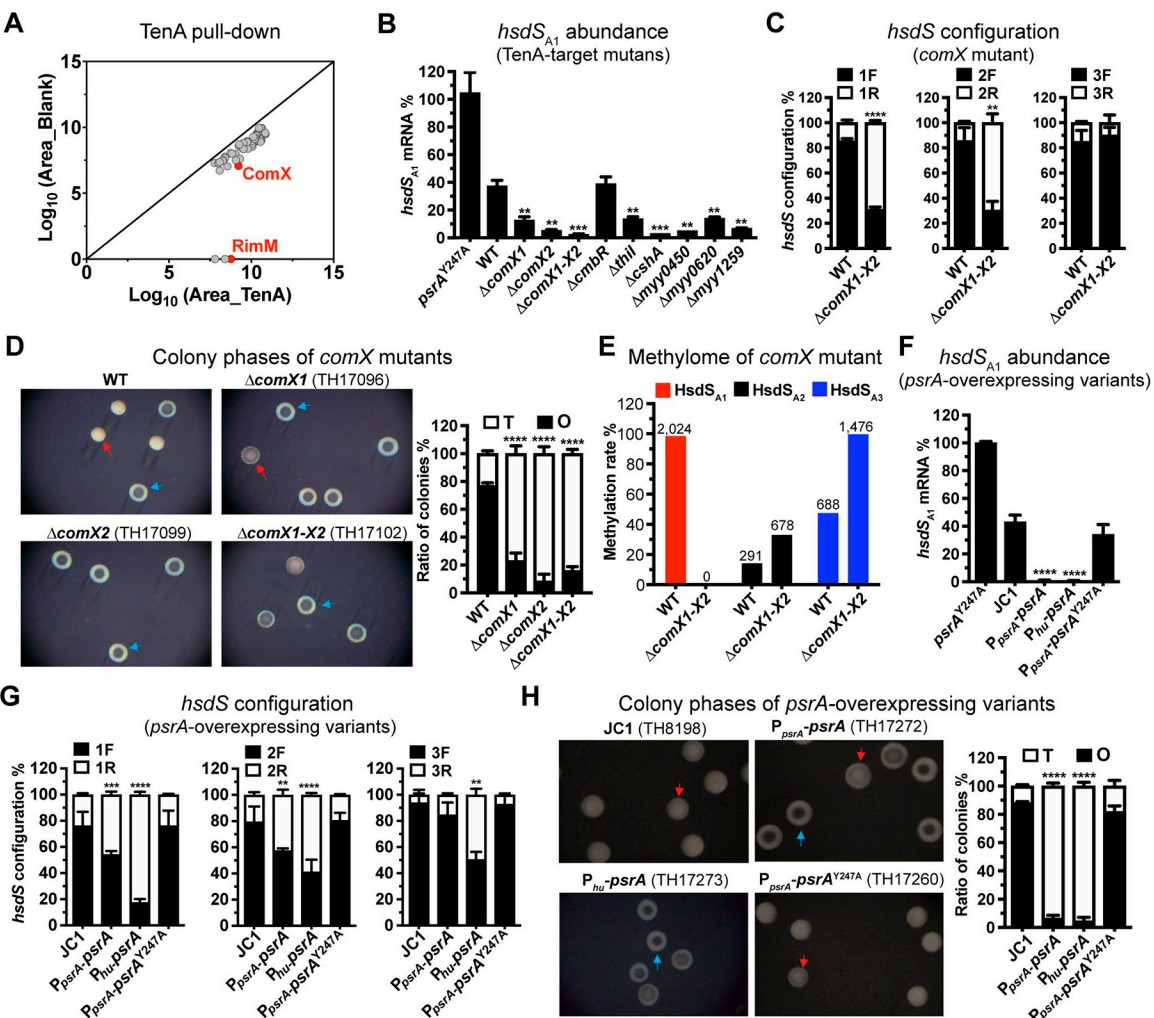

**Fig 5. Modulation of inversion-driven epigenetic and colony phase switch by TenA-associated ComX. (A)** TenA-associated proteins enriched by affinity pull down. Solid line represents the threshold of enrichment by rTenA compared to the negative control (Blank). **(B)** Relative abundance of the $hsdS_{A1}$ mRNAs in the deletion mutants of TenA-associated protein. **(C)** The $hsdS$ allelic configurations of *comX* mutant were detected and shown as in Fig 2E. **(D)** Colony phenotype and ratio between O and T colonies of the *comX* mutants. **(E)** The genomic DNA methylation state of the *comX* mutant was detected and shown as in Fig 3B. Number of methylated motifs by SMRT sequencing detection was labelled on the top of each single bar. **(F to H)**, Relative abundance of the $hsdS_{A1}$ mRNA (F), the $hsdS$ allelic configurations (G), colony phenotype and ratio between O and T colonies (H) of *psrA*-overexpressed variants were detected and shown as in (B), (C) and (D), respectively.

of *comX* (**Fig 5E**). Taken together, these results indicated that both copies of ComX are required for regulating the *hsdS* inversions independent of the competence state.

Based on the known function of ComX as a transcriptional regulator, we determined how ComX and TenA are functionally linked to regulate *hsdS* inversions by comparing the transcriptomes of the *tenA*-overexpressed (P*spxA1*-*tenA*) and *comX* deletion (Δ*comX1-X2*) strains using RNA sequencing. As compared with the parental strain, a total of 33 and 50 differentially expressed genes with a 1.5-fold change of transcripts were identified for P*spxA1*-*tenA* and Δ*comX1-X2*, respectively (**S5 Table**). These results showed that *psrA* was the only upregulated gene in these two mutants (**Table 1** and **S6B Fig**). To ascertain if enhanced expression of PsrA alters the orientation of *hsdS* inversions, we generated two *psrA*-overexpressed strains, in which *psrA* was duplicated in the *bgaA* locus under the control of either the *psrA* (P*psrA*-*psrA*)

**Table 1. Differentially expressed genes in the P$_{spxA1}$-*tenA* and Δ*comX1-X2*.**

| Gene_ID | Functional description | Fold change of transcripts* | | P value[†] |
|---------|------------------------|----------|----------|-----------|
| | | P$_{spxA1}$-*tenA* | Δ*comX1-X2* | |
| *myy02560* | PsrA, invertase | 1.5 | 1.6 | 0.0 |
| *myy0248* | RibH, riboflavin synthase | -1.5 | -2.2 | 0.0 |
| *myy0250* | RibB, riboflavin synthase | -1.5 | -2.3 | 0.0 |
| *myy0251* | RibD, riboflavin synthase | -1.7 | -2.3 | 0.0 |
| *myy1793* | RafG, sugar ABC transporter permease | -1.7 | -1.5 | 0.0 |
| *myy2067* | ArcA, arginine deiminase | -1.8 | -1.5 | 0.0 |
| *myy2068* | ArcB, ornithine carbamoyltransferase | -1.8 | -1.6 | 0.0 |

*Fold change in gene expression as assessed by RNA-seq.

[†]Adjusted *P* values, *P* values were adjusted using the Benjamini and Hochberg method.

or *hu* promoter (P$_{hu}$-*psrA*). The P$_{psrA}$-*psrA* and P$_{hu}$-*psrA* strains showed 2.5- and 3.7-fold increase in *psrA* mRNA, respectively (**S6C Fig**). Surprisingly, both strains exhibited dramatic reduction in the *hsdS*$_{A1}$-carrying variant in the clonal populations (**Fig 5F**), which was further confirmed by inverted repeat-based qPCR result (**Fig 5G**). Consistently, both the *psrA*-overexpressed variants dramatically lost the capacity of O colony formation (**Fig 5H**). This phenotypic impact of *psrA* overexpression was not observed in the strain with an inactive *psrA*. These results suggested that TenA indirectly regulates *hsdS* inversions by ComX-mediated transcriptional control of *psrA*.

We also verified the functional impact of the riboflavin biosynthesis locus on *hsdS* inversions, because the expression of multiple *rib* genes (*ribH*, *ribB* and *ribD*) in this locus was downregulated in both the P$_{spxA1}$-*tenA* and Δ*comX1-X2* strains (**Table 1** and **S6B Fig**). Deleting the whole *rib* locus (*ribH*, *ribA*, *ribB* and *ribD*) led to significant reduction in *hsdS*$_{A1}$-carrying bacterial ratio (**S6D Fig**) and O colony formation (**S6E Fig**). Consistently, the *hsdS* gene configurations in the *cod* locus also displayed as the T phase direction in the Δ*rib* mutant (**S6F Fig**). Since riboflavin is broadly involved in cellular metabolism [39], this result suggests that pneumococcal *hsdS* configuration and methylome are intertwined with cellular metabolism.

## TenA targets ribosomal chaperonin RimM to control the abundance of ComX

The overwhelming enrichment of ribosome-related proteins among the TenA-associated protein hits suggests that TenA physically interacts with the ribosome. We selectively characterized the potential interaction of TenA with RimM, the most enriched ribosomal protein, by bacterial adenylate cyclase-based two-hybrid (BACTH) system. As depicted in **Fig 6A**, TenA showed strong binding interaction with RimM, but not the *hsdS* inversion-modulating protein ComX. RimM has been reported to be involved in the maturation of the ribosomal 30S subunit by binding to ribosomal protein S19 [40,41]. As the BACTH results indicated an interaction of TenA with RimM but not with ComX, we hypothesized that TenA acts on RimM to control ComX abundance and thereby *psrA* expression. Hence, we assessed the impact of RimM on *hsdS* inversions by constructing the *rimM* deletion mutant. Although deletion of *rimM* led to severe growth defect that prevented characterization of colony phenotype, the *rimM* mutant showed a significant reduction in the *hsdS*$_{A1}$-carrying variant in the clonal population (**Fig 6B**). Supporting our hypothesis, these results indicate that RimM is involved in modulating *hsdS* inversions.

To further understand how TenA interacts with RimM, we predicted the structure of the molecular complex using AlphaFold3 [36]. The bioinformatic analysis strongly suggested that

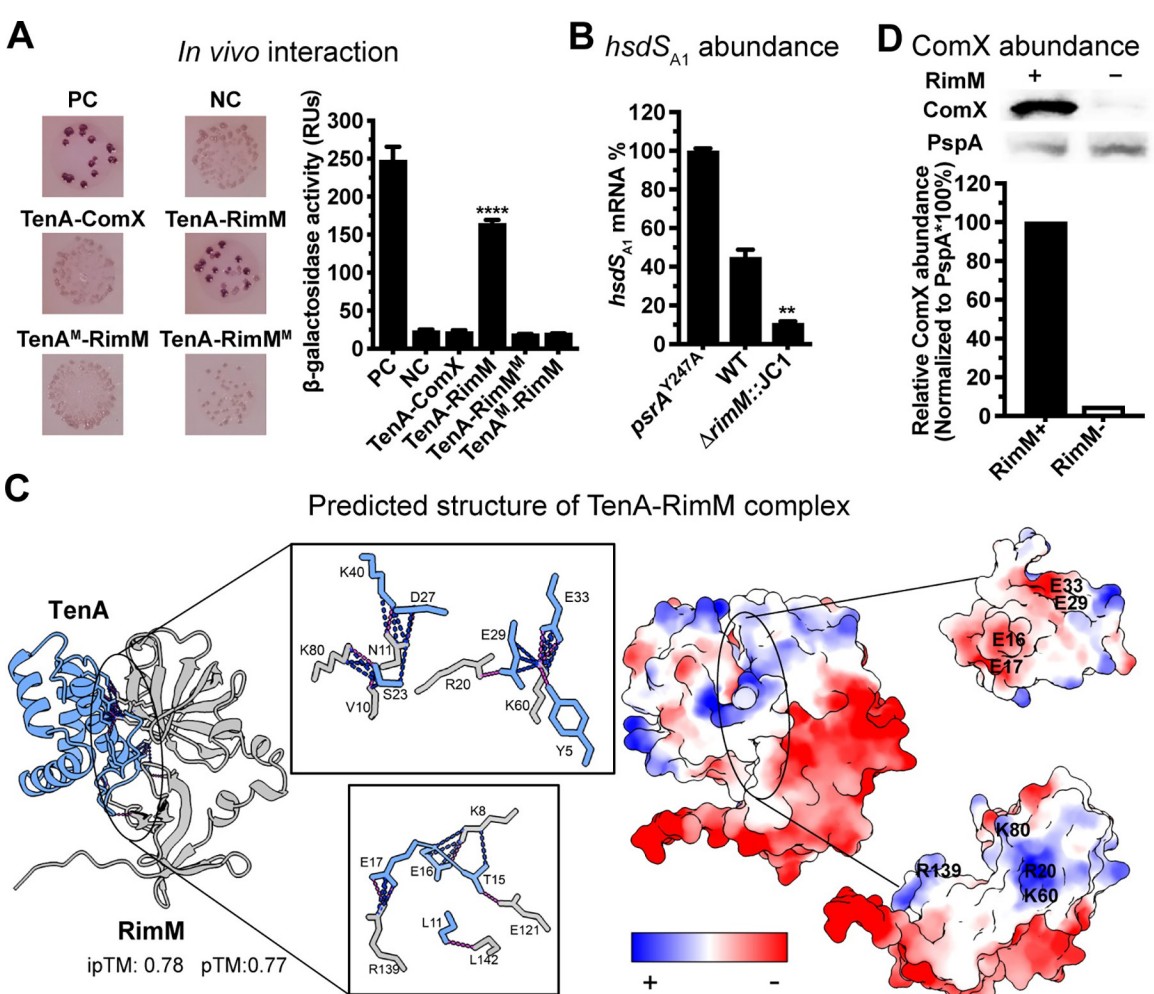

**Fig 6. Requirement of RimM for the abundance control of ComX by TenA.** (**A**) Detection of TenA interactions with potential targets by bacterial two-hybrid (BACTH) assay. Colonies on the MacConkey/maltose plates (left panel) and β-galactosidase activity (right panel) are shown for each reporter strain. PC, positive control (pKT25-*zip* and pUT18C-*zip*). NC, negative control (empty vectors pKT25 and pUT18C). The mean ± s.d. of three values (from three individual experiments) of each strain is presented in a single bar. (**B**) Relative abundance of the $hsdS_{A1}$ mRNA in the *rimM* mutant. (**C**) Predicted interaction between TenA and RimM. The structural models and surface electro statics were predicted using AlphaFold3 and drawn in ChimeraX. The model confidence values of each model are calculated and labelled with ipTM and pTM score. The predicted surface hydrogen bonds and contacts of TenA (light blue) and RimM (grey) are indicated with purple dash line and blue dash line in the enlarged frame, respectively. The surface electro statics are indicated with red (negatively charged) and blue (positively charged). (**D**) Relative abundance of ComX in the *rimM* mutant. The protein band and relative amounts of ComX (~19.9 kDa) in strains Δ*clpP* and Δ*clpP*-Δ*rimM*::JC1 were determined by western blotting using an anti-His antibody. For detection, ClpP was pre-deleted to prevent ComX from being degraded by protease. PspA, pneumococcal surface protein A (~80.1 kDa), was used as an internal loading control and to normalize the abundance of ComX. The intensities of each band on the chemiluminescence films were quantified using ImageJ.

TenA and RimM form a strong molecular complex (**Figs 6C** and **S7**). As shown in **Fig 6C**, the predicted complex exhibited a total of 68 potential contacts containing 13 hydrogen bonds. The positively charged interface of RimM may interact with the negatively charged interface of TenA. The potential interface consists of four glutamic acids at positions 16, 17, 29 and 33 of TenA, and four basic amino acids at positions R20, R139, K60 and K80 of RimM. Mutating all of these predicted binding sites of TenA (TenA^M) and RimM (RimM^M) by alanine walk can relieve their interaction in BACTH system (**Fig 6A**). In the context of the TenA-ComX functional linkage, this result supports the notion that TenA indirectly regulates the orientation of

*hsdS* inversions by altering the protein abundance of ComX via interacting with RimM. To test this hypothesis, we quantified the abundance of ComX in WT (RimM$^+$) and Δ*rimM*::JC1 (RimM$^-$) variants in the absence of serine protease ClpP, since ClpP has been reported to degrade ComX [38]. The immunoblotting result revealed a comparable abundance of PspA, a cell-wall associated protein used as the internal control, in both the Δ*clpP* and Δ*clpP*-Δ*rimM*::JC1 strains. Conversely, ComX was abundant only in Δ*clpP* but not Δ*clpP*-Δ*rimM*::JC1 (**Fig 6D**). This result suggests that TenA targets RimM to control the abundance of ComX.

## The AT-rich region upstream of the *spxA1* promoter controls the expression and activity of the *spxA1-tenA* operon

To understand the biological importance of the 5'-ATAT-3' insertion that boosts the expression of the type II toxin TenA (**Fig 1C**), we characterized the 209-bp upstream sequence of *spxA1* from 33,625 genome sequences of *S. pneumoniae* in the PubMLST database (https://pubmlst.org) (**S6 Table**) [42]. As shown in **Fig 7A**, the WebLogo [43] analysis revealed a long stretch of perfect microsatellite repeats (AT-rich region) is situated between an imperfect palindromic sequence (5'-TTTRC-AT$_n$-CRTTT-3') in all of these sequences, in which R is A or G. More importantly, there are extensive variations in the length of the AT-rich region among these genomes, ranging from 10 bp to 38 bp (**Fig 7B**). 93.3% of these genomes contain an 18-bp AT di-nucleotides sequence in this region (31,358 hits), including the laboratory strains D39, TIGR4 and ST556. The spontaneous mutant of ST556 Δ*rr06*$^{rev-N}$ (TH9551) harbors the fourth most abundant AT-rich variant with 22-bp length (281 hits) (**Fig 1B**). All of these sequences of AT-rich region in pneumococcal strains are almost always occupied by a single or multiple copies of the AT di-nucleotides, which resembles the variable tandem TA di-nucleotide repeats upstream of the promoter sequence of the pilus gene cluster in *Haemophilus influenzae* [44].

We next determined the potential impact of the length of the AT-rich region on the expression of *tenA* by replacing the AT-rich sequence in ST606 (18 bp) with representative variants with various lengths (10, 22 or 38 bp). The qRT-PCR results showed that the mRNA levels of *tenA* increased by 1.5- and 1.6-fold in strains containing the 22- and 38-bp AT-rich sequences, respectively, while 10-bp AT-rich counterpart showed comparable *tenA* transcripts with that of the parental strain (**Fig 7C**). In a similar manner, the isogenic strains with the 22- and 38-bp AT-rich sequences in the *spxA1* promoter region showed a significantly reduced proportion of *hsdS*$_{A1}$-carrying subpopulations (**Fig 7D**). As compared with the parental strain that possessed *hsdS*$_{A1}$-dominant allele (33.0% of *hsdS*$_{A1}$ mRNA), both strains containing 22- and 38-bp AT-rich sequences generated 4.7% *hsdS*$_{A1}$ transcripts. Consistent with the reduction of the *hsdS*$_{A1}$-carrying bacteria, the O colonies of these strains were reduced from 78.0% in the parental strain to 15.2% and 14.9% in the AT-22 and -38 variants, respectively (**Fig 7E**). The strain containing the 10-bp AT-rich sequence showed comparable levels in both the *hsdS*$_{A1}$ mRNA and O colony as those of the parental strain (**Fig 7D** and **7E**). The dramatic impact of the spontaneous variation in the AT-rich region on the transcriptional regulation of TenA, and thus the resultant *hsdS* inversions and colony phase indicated that *S. pneumoniae* employs the sequence variation in the AT-rich region to generate phenotypic diversity in clonal populations through modulating the expression of the SpxA1-TenA toxin-antitoxin system.

Taken together, we propose a working model of the SpxA1-TenA toxin-antitoxin system (**Fig 7F**). Spontaneous sequence variations of the AT-rich region in the *spxA1* promoter region alter the transcription of the *spxA1-tenA* operon and the unbalanced functional relationship between SpxA1 and TenA; the excess toxin TenA targets the ribosomes by interacting with ribosome biogenesis chaperone RimM to exert its "toxicity", somehow reducing the cellular

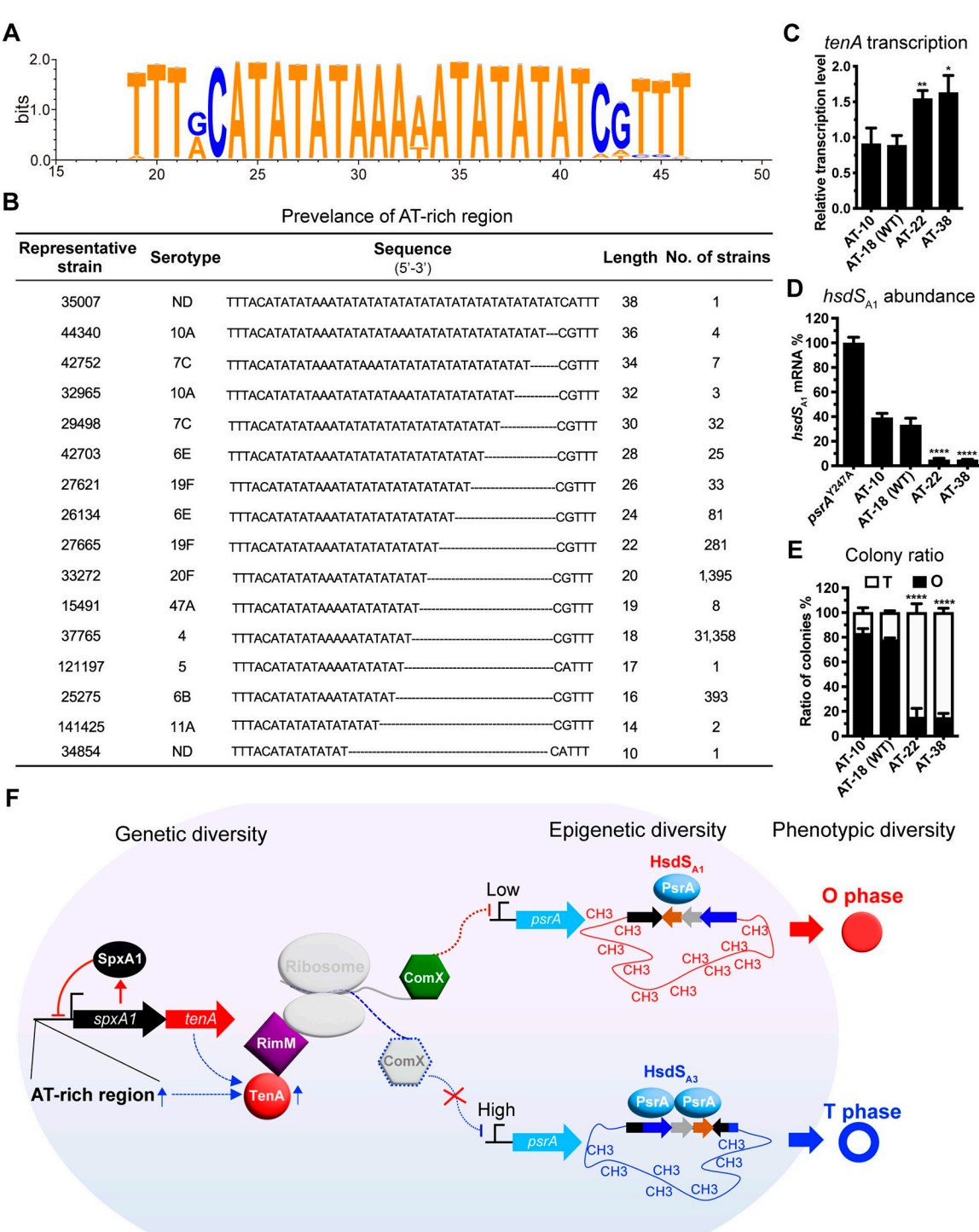

**Fig 7. Functional linkage of an AT-rich region with TenA-modulated epigenetic and colony phase switch.** (**A**) The conservation of AT-rich sequence in the promoter region of *spxA1-tenA* was illustrated by WebLogo (https://weblogo.threeplusone.com). (**B**) Prevalence and length of AT-rich regions in 33,625 isolates from the genome database of *S. pneumoniae* PubMLST (https://pubmlst. org). Each length of AT-rich region is enumerated with a representing sequence from different pneumococcal isolate. (**C** to **E**) Relative expression of *tenA* (C), relative abundance of the *hsdS*$_{A1}$ mRNA (D) and colony ratio (E) of ST556 variants with different lengths of AT-rich sequence. (**F**) A model depicting our view of pneumococcal epigenetic and colony phase variation controlled by the SpxA1-TenA system.

abundance of ComX; the reduced ComX leads to the enhanced expression of invertase PsrA and thereby shift the *hsdS* inversions in the *cod* locus toward non-*hsdS*$_{A1}$ allelic configurations and methylomes; the resulting epigenetic variations yield phenotypical diversity in colony morphology and potentially other traits by an uncharacterized mechanism. In short, this spontaneous genetic variation in the expression of the SpxA1-TenA toxin-antitoxin system may represent a programmed mechanism of epigenetic adaptation in *S. pneumoniae*.

## Discussion

### TenA and SpxA1 form a new type II TA system

Led by a serendipitous observation in our previous work [11], we found that the modest over-expression of *tenA* is responsible for the striking disappearance of opaque colonies in the *rr06* mutant. The subsequent experiments provided multiple lines of evidence that TenA is the toxin component of the previously uncharacterized type II TA system, with SpxA1 as the antitoxin partner. First, the sequence features of the *spxA1* and *tenA* gene locus fit well with the typical genetic organization of type II TA systems [18]. The *spxA1* antitoxin gene is placed in front of the *tenA* toxin gene overlapping coding sequences. In addition, the protein products of SpxA1 (133 amino acids) and TenA (92 residues) are reminiscent of the small polypeptide sizes of type II TA systems. Second, our transcriptional analysis showed that *spxA1* and *tenA* are co-transcribed as an operon, which is consistent with the operonic feature of the type II TA genes. Third, TenA exhibited a toxic effect on *S. pneumoniae* (enhanced autolysis) when overexpressed. This toxic activity agrees with the TenA-induced growth inhibition in *E. coli*. The relatively milder toxic effect of TenA in *S. pneumoniae* is consistent with the mild toxicity of other type II pneumococcal TA pairs. As an example, the PezAT TA system does not affect the growth of strain R6, but is associated with autolysis at the stationary phase and transformability [19,45]. Lastly, autoregulation of the *spxA1-tenA* operon by SpxA1 is another feature of type II TA systems. This is characterized by transcriptional overexpression of *tenA* and the associated phenotypes in the *spxA1* mutant.

### SpxA1 autoregulates the expression of the *spxA1-tenA* operon

A major feature of the type II TA system is its autoregulation at the transcriptional level [18]. This regulation mostly occurs by binding antitoxins alone or the toxin-antitoxin complex to the promoters of the TA encoding operons to repress the transcription [35,46]. Consistent with the anticipated binding interaction between SpxA1 and TenA, AlphaFold3 analysis predicted the formation of a SpxA1-TenA complex. It is expected that the SpxA1-TenA complex would interact with the 5' non-coding sequence of the *spxA1-tenA* operon to repress the transcription of the TA system. Potential proteolysis of SpxA1 under certain conditions would lead to de-repression of the operon and unleash the toxic effect of TenA. Alternatively, SpxA1 alone may exert its transcriptional regulation of the *spxA1-tenA* operon by interacting with RNA polymerase at the promoter sequence. Spx, the SpxA1 homolog in *B. subtilis*, represses the transcription of many stress response-associated genes by interacting with the α subunit of RNA polymerase and the promoter sequences under physiological conditions [47–50], but it is selectively degraded by ATP-dependent protease ClpXP under stress conditions [47,51]. The SpxA1 homologs in numerous Gram-positive bacteria have also been shown to regulate the genes associated with stress responses [52–54]. It is noteworthy that our TenA affinity pull-down did not identify SpxA1. This might be due to excessive degradation of SpxA1 by an unknown protease under the excess of TenA. It is also possible that SpxA1 neutralizes the toxic activity of TenA without forming a typical toxin-antitoxin complex.

## TenA regulates pneumococcal epigenome by modulating the *hsdS* inversions in the *cod* locus

The striking effect of TenA overexpression on pneumococcal colony phase has enabled us to find the functional linkage between TenA and the DNA configurations in the *cod* locus. The overexpression of *tenA* in the 5'-ATAT-3' insertion strain led to a dramatic overhaul of the *hsdS* allelic configurations. This genetic shift is characterized by loss of the $hsdS_{A1}$ allele and simultaneous enrichment of $hsdS_{A2}$ and $hsdS_{A3}$ allelic variants. The TenA-driven allelic loss of $hsdS_{A1}$ was repeatedly verified under various *tenA*-overexpressed settings, such as ectopic expression of *tenA* under the control of *spxA1* or *hu* promoter in the *bgaA* locus, deletion of *spxA1*, and sequence changes in the AT-rich region. At the epigenetic level, the pneumococcal strain with the *hu* promoter-driven *tenA* completely lost the 6-mA methylation of the DNA motif recognized by the $HsdS_{A1}$-specific DNA methyltransferase, with more abundant methylation of $HsdS_{A2}$- and $HsdS_{A3}$-recognized sequences.

## TenA exerts profound epigenetic impact on pneumococcal physiology

The TenA-driven epigenomic overhaul dramatically alters various aspects of bacterial biology. This was manifested by overwhelming switch of pneumococcal colony phase from the opaque colony phase in WT strain to the transparent counterpart under multiple genetic settings in which *tenA* was overexpressed. While the mechanistic link between genome methylation pattern and colony phase remains to be defined, it is certain that the methylome determined by individual $HsdS_A$ shapes pneumococcal colony opacity via complex processes. A large body of literature has documented significant differences between the opaque and transparent colony phases in the amount of capsular polysaccharide [4,55], cell wall polysaccharide [2,4,56], multiple proteins [57–59] and fatty acids [60]. Moreover, the O and T colony phases also differ in autolysin LytA-driven cell lysis (autolysis) [3,61] and natural genetic transformation [62]. In this context, future investigations are warranted to define how TenA exerts its epigenetic impact on pneumococcal pathobiology.

## ComX functionally links TenA and *hsdS* inversions

ComX is known for its essential role in transcriptional activation of the late competence genes for DNA uptake and processing [13,15,16]. This study has demonstrated that ComX is required for the stabilization of the O phase-defining $hsdS_{A1}$ allele in the *cod* locus. Deleting one or both copies of *comX* led to the loss of the $hsdS_{A1}$ allele in the *cod* locus. The *comX1-X2* mutant completely lost the 6-mA methylation of all $HsdS_{A1}$-specific motif loci in the pneumococcal genome. These genetic and epigenetic features were consistent with the dramatic shift of the colony phenotype toward the T phase in these mutants. Mechanistically, our affinity pull-down experiments suggested that ComX links TenA with the *hsdS* inversions by repressing the expression of the *psrA* invertase gene. Using TenA as bait led to the identification of ComX as one of the TenA-binding protein candidates. Since the two-hybrid test did not detect direct binding interaction between the two proteins, it appears that TenA and ComX cooperate in regulating *hsdS* inversions without direct binding to each other by an unknown mechanism.

The dramatic impact of single *comX1* or *comX2* deletion on *hsdS* inversions indicates that the inversion machinery is highly sensitive to the cellular level of ComX. This provides a functional explanation for the fact that all sequenced genomes of *S. pneumoniae* possess both *comX1* and *comX2*. Although ComX is induced by the two-component system ComDE in the competence state [63], our previous work has shown that deleting *comE* in *S. pneumoniae* does

not affect pneumococcal methylome or colony phase [11]. These lines of information indicated that the basal level of ComX under non-competence conditions is sufficient to regulate *hsdS* inversions and methylome. Consistently, ComW regulated by the ComDE two-component system is also involved in regulating *hsdS* inversions and methylome [11]. Since ComX and ComW functionally cooperate to activate the expression of the late competence genes by interacting with RNA polymerase and the conserved "*com* box" sequence, it is possible that the two competence-associated proteins also work together to regulate *hsdS* inversions by repressing the transcription of *psrA*. Since our previous work suggests that the relatively higher levels of the PsrA invertase favor the *in vitro* inversions that generate the non-$hsdS_{A1}$ allelic variants [64], reduced ComX would favor the non-$hsdS_{A1}$ allelic variants and the resulting methylome.

## TenA may target pneumococcal protein synthesis to exert its regulatory role

This notion is based on multiple lines of experimental evidence in this work. In particular, the vast majority of the protein hits in the TenA affinity pull-down were associated with protein synthesis. In the context of specific binding between TenA and RimM (see below), this finding suggested that TenA exerts its regulatory role by directly interacting with RimM. Alternatively, TenA may also indirectly interact with other ribosomal proteins via RimM as a molecular bridge. Moreover, our bacterial two-hybrid assay revealed binding interactions between TenA and RimM, which was supported by the highly probable complex of the two proteins predicted by AlphaFold3. Since RimM is crucial for the maturation of the small ribosomal subunit [41], TenA binding interaction with RimM may interfere with ribosomal maturation and thereby protein synthesis, which leads to a decrease in the cellular abundance of ComX and other proteins that are associated with *hsdS* inversions. The selective decrease of ComX in the *rimM* mutant further suggests that TenA preferentially affects the synthesis of ComX and probably other *hsdS*-associated proteins by an unknown mechanism.

## Sequence variations in the *spxA1* promoter region control the functional balance of the SpxA1-TenA TA system and thereby the epigenetic landscape of *S. pneumoniae*

The sequence variations in the AT-rich region upstream of the *spxA1* promoter among pneumococcal strains are reminiscent of the TA repeat-driven variation in piliation of *H. influenzae* [44]. Variations in TA di-nucleotide repeats upstream of the *hif* pilus gene promoter sequence control the level of pilus in *H. influenzae* [44]. While the sequence with 10 TA repeats drives maximal transcription of the *hif* genes and pilus production, those carrying 9 or 11 TA repeats produce reduced *hif* genes transcription and undetectable pilus. In this study, we identified the first case in which *S. pneumoniae* use a similar AT repeat variation in the promoter region to control the balance of the SpxA1-TenA TA system. The AT-rich sequence with certain lengths (e.g., 10 and 18 bp) drives relatively lower level of *tenA* transcription and higher levels of $hsdS_{A1}$-carrying bacteria, whereas the sequences with other lengths (e.g., 22 and 38 bp) yield the opposite outcomes. While this type of repeat sequence variations is explained by the slipped strand mispairing during DNA replication [65], recent single-cell RNA sequencing in *Klebsiella pneumoniae* has revealed that spontaneous expression of certain transposases is responsible for insertion or deletion of repeat sequences in the genome and leads to intra-population heterogeneity in gene expression and phenotype [66].

In the context of the toxin-antitoxin system, intra-population variations in the number of AT repeats define the transcriptional expression level of *tenA* and the downstream molecular events. When the operon is controlled by the low-transcription AT variants, SpxA1 neutralizes the toxic effect of TenA by forming a protein complex under the physiological conditions, which

would enhance the synthesis of ComX and other *hsdS*-associated proteins, and the formation of the $hsdS_{A1}$ allelic configuration and the corresponding methylome. When the AT-rich region takes on the high-transcription sequences, the toxic activity of TenA is unleashed by binding to RimM (and/or other target) and inhibiting protein synthesis. The reduced level of ComX leads to transcriptional increase of *psrA* and an allelic shift of *hsdS* inversions away from the $hsdS_{A1}$ allelic configuration and corresponding methylome. Phenotypically, the number of AT repeats upstream of the *spxA1* promoter broadly defines the cellular metabolism and structure of *S. pneumoniae* as manifested by variations in the amount of capsular polysaccharide, teichoic acid, fatty acids and proteins, and in the cellular behavior (e.g., autolysis and colony opacity).

## Methods

### Bacterial strains, cultivation and chemical reagents

All experimental strains used in this study are listed in **S7 Table**. All pneumococcal strains were cultivated in Todd-Hewitt broth with 5% yeast extract (THY), tryptic soy broth (TSB), chemically defined medium with yeast extract (C+Y) or on tryptic soy agar plate (TSA) with 5% sheep blood at 37˚C as described [11,67]. Streptomycin (150 μg/ml) and kanamycin (400 μg/ml) were added to the medium when necessary. *E. coli* strains DH5α and BL21-DE3 were used for subcloning of plasmids and protein expression, respectively. *E. coli* strains were grown in Luria-Bertani (LB) broth or on LB agar plates with appropriate concentrations of antibiotics (50 μg/ml kanamycin, 100 μg/ml ampicillin or 20 μg/ml chloramphenicol). Primers were synthesized by Ruibiotech (Beijing, China) and are listed in **S8 Table**. All chemical reagents were purchased from Sigma (Shanghai, China) unless otherwise noted. DNA processing enzymes were purchased from New England Biolabs (Beijing, China). Sanger sequencing data were obtained from Ruibiotech (Beijing, China).

### Strain construction

Pneumococcal mutagenesis was carried out in streptomycin-resistant strains ST606 (ST556 derivative, serotype 19F), TH6671 (P384 derivative, serotype 6A) and TH6675 (ST877 derivative, serotype 35B) essentially as described [9]. The mutant construction procedures are listed in **S9 Table**. All unmarked deletion mutants in *S. pneumoniae* were constructed by Janus cassette (JC1)-mediated counter selection in two steps [9]. Briefly, JC1 (a modified Janus cassette) was amplified from *S. pneumoniae* ST588 with primers Pr1098 and Pr9840 [9], digested with XbaI/XhoI, and ligated with the up- and down-stream sequences. The products were introduced into target strains by natural transformation for homologous recombination and sequence replacement; transformants were selected for resistance to kanamycin and screened for sensitivity to streptomycin. Subsequently, unmarked gene deletion mutants were obtained by transformation with either ligation or fusion PCR products of the up- and down-stream sequences to replace JC1. The isogenic revertants were constructed by using the amplicon of the wild-type sequence to replace JC1. The gene overexpressed variants were generated similarly as above in the TH8198 (Δ*bgaA*::JC1) background. The JC1 background strain was transformed with a fusion PCR product consisting of the flanking regions of *bgaA*, the promoter sequences of the *spxA1*, *psrA* or *hu* with the target genes respectively, followed by selection with streptomycin. All mutations were verified by PCR amplification and DNA sequencing.

### Microscopic assessment of bacterial colonies

Observation and quantification of pneumococcal colony opacity were carried out on TSA plates supplemented with catalase as described [68]. Briefly, bacteria were cultivated in THY

medium to an $OD_{620\ nm}$ of 0.5 at 37°C with 5% $CO_2$. The bacterial suspension was then diluted with Ringer's solution and separately spread on 9-cm-diameter TSA plates with 6,000 units catalase. After incubation at 37°C with 5% $CO_2$ for 17 hours, the representative colonies of each strain on the catalase-TSA plate were photographed under a dissection microscope. Meanwhile, the ratio between the two types of colonies for each strain was obtained with triplicate plates each time, and subsequently repeated at least twice.

## RNA isolation and sequencing

RNA sequencing (RNA-seq) was carried out as described with minor modifications [11]. In brief, bacteria were collected from the colonies on catalase-TSA plates at the indicated times, washed once with pre-chilled Ringer's solution and frozen in liquid nitrogen. Total RNA was extracted from the frozen samples with the RNApure Bacteria kit (CoWin Biotech, China) and further purified with RNeasy Protect bacterial kit (Qiagen, Germany) according to the manufacturer's instructions. RNA-seq was performed at the Novogene Bioinformatics Technology (Tianjin, China). In brief, the ribosomal RNA was removed from total RNA and then precipitated with ethanol. After fragmentation, the paired-end library was further constructed and sequenced with Illumina NovaSeq PE150 platform. The resulting raw reads of fastq format were further processed through fastp software and then mapped to the genome of *S. pneumoniae* ST556 (accession CP003357.2) using Bowtie 2.3.1 and Tophat 2.1.1. The FeatureCounts was used to count the reads numbers mapped to each gene (Transcripts Per Kilobase Million, TPM). Transcript abundance files were then processed in the DESeq2 R package [69]. The resulting *P* values were adjusted using Benjamini and Hochberg's approach for controlling the false-discovery rate. Significant difference was defined by an at least a 1.5-fold change and a Padj less than 0.05. All sequence data have been deposited in the NCBI Gene Expression Omnibus database under the following accessions: SRR29843789 (ST606, WT), SRR29843788 (TH17102, Δ*comX1-X2*), SRR29843787 (TH8198, Δ*bagA*::JC1) and SRR29843786 (TH14122, $P_{spxA1}$-*tenA*). The data of each sample represents the means of three independent experiments.

## mRNA abundance quantification by qRT-PCR

Quantitative real-time reverse transcriptase PCR (qRT-PCR) was performed to quantify the relative transcriptional levels of target genes and $hsdS_{A1}$ allele abundance as described [11,26]. In brief, 1 μg of total RNA was extracted as described for RNA sequencing and used for cDNA synthesis following the instruction of iScript cDNA synthesis kit (Bio-Rad, USA). The *era* gene was amplified with primer pair Pr7932/7933 as the internal control to normalize to the expression of target genes by the comparative threshold cycle method.

For the $hsdS_{A1}$ allele quantification, the 367-bp $hsdS_{A1}$ allele-specific sequence was quantified to assess the relative abundance of $hsdS_{A1}$ allelic gene in the clonal population [11]. The qRT-PCR was performed with primer pairs Pr16174/16175 and Pr16178/16179 to detect the specific sequence in the $hsdS_{A1}$ allele and the 5' non-invertible region (267 bp) shared by the six $hsdS_A$ alleles as an internal reference, respectively. The average $C_T$ value of $hsdS_{A1}$ allele was first normalized to the mean $C_T$ value of the $hsdS_A$ alleles internal reference in each strain, and further normalized to the $hsdS_{A1}$-fixed mutant $psrA^{Y247A}$ by subtracting the average $\Delta C_T$ value of $psrA^{Y247A}$ from that of each strain. The relative abundance of $hsdS_{A1}$ mRNA of each strain is presented as $(2^{-\Delta\Delta CT})$% given that the relative abundance of $hsdS_{A1}$ mRNA in $psrA^{Y247A}$ is 100%. The data from one representative experiment are presented as the mean value of triplicate samples ± the standard deviation (s.d.) for each strain. Each experiment was repeated at least three times. The primers used to amplify the target genes are listed in **S8 Table**.

## *hsdS* allelic configuration quantification by qPCR

The *hsdS* inversion frequency mediated by each inverted repeat indicates corresponding *hsdS* allelic configuration. IR-mediated *hsdS* inversion frequency was detected by quantitative PCR (qPCR) using genomic DNA as template essentially as described [10]. The genomic DNA of each strain was extracted from bacteria collected on catalase-TSA plates using a TIANamp bacterial DNA kit (Tiangen, China) according to the manufacturer's protocol. The primers used to measure the fractions of either forward (F) or reverse (R) orientation are labelled as in **S2 Fig** and listed in **S8 Table**. The PCR fragments representing "forward" and "reverse" orientation of IR1-, IR2- and IR3-inversion were amplified by corresponding primers, respectively. All the orientations of IR1, IR2 and IR3 in the O phase $hsdS_{A1}$ gene configuration as the "forward state", in contrast, the T phase $hsdS_{A2-6}$ gene configuration as the "reverse state". For the "forward" state of IR1, IR2 and IR3-mediated inversion, the specific fragment can be amplified by the primer pair P1/P3, P4/P6 and P7/P9, respectively. The reversed specific fragments were detected by P1/P2 (IR1), P4/P5 (IR2) and P7/P8 (IR3), respectively. qPCR mix was carried out with 10 ng genomic DNA per 25 μl-reaction as template and following the manufacturer's protocol of iTaq Universe SYBR Green Supermix (Bio-Rad, USA) or MagicSYBR Mixture (CoWin Biotech, China). To calculate the relative forward and reverse configurations of the *hsdS* sequences, the average $C_T$ value of each inverted repeat was first normalized by subtracting the $C_T$ value of the internal reference gene *era*. The final inversion frequency was defined by the equation $2^{-(\Delta CT)}$ of each direction/total $2^{-(\Delta CT)}$ value of forward and reverse × 100%. Relative composition of both orientations in a single population is shown as mean ± s.d. of a representative experiment. Each experiment was replicated independently at least twice.

## SMRT sequencing and methylome analysis

Pneumococcal methylome analysis was assessed by the single molecule real-time (SMRT) sequencing essentially as described [8]. Genomic DNA was prepared as described for qPCR and sequenced in the Novogene Bioinformatics Technology (Beijing, China). Briefly, 10Kb-SMRT Bell library was constructed and sequenced using the PacBio RSII sequencing platform. The low-quality reads were removed by the SMRT Link v5.0.1 and the filtered reads were subsequently processed by SMRT Portal (Version 2.3.0) to generate one contig without gaps. The raw data representing of the SMRT sequencing are available at the NCBI database under the following accessions: SRR29849704 (ST606, WT), SRR29849701 (TH15070, P*hu*-*tenA*), SRR29849703 (TH17102, Δ*comX1-X2*) and SRR29849700 (TH9551, Δ*rr06*^rev-N^).

## Assessment of TenA toxicity

To induce the expression of *spxA1* or *tenA* in *E. coli* BL21-DE3, the *spxA1* or *tenA* genes were cloned in a shuttle vector pIB166 and transcribed under the control of an IPTG-inducible P*lac* promoter [70,71]. Briefly, *spxA1* and *tenA* were amplified, fused with the promoter sequence of P*lac*, digested and cloned into the ApaI and XhoI sites of the pIB166 vector. To detect the impact of TenA on bacterial growth, *E. coli* strains containing IPTG-inducible plasmids were first cultured in LB broth with chloramphenicol to an $OD_{600 \text{ nm}}$ of 0.6. After dilution with PBS, 10 μl-bacterial suspension were spotted onto the LB agar plate supplemented with 1 mM IPTG and cultured for about 24 hours at 37°C. For the measurement of growth curve, *E. coli* and pneumococci were reinoculated into fresh LB (with 1 mM IPTG) or C+Y medium after 1:50 dilution, respectively, and were incubated at 37°C for 24 hours in 48-well flat bottom plates (Costar, China). The optical density was monitored every two hours using a Tecan Infinite F200 Pro microtiter plate reader (Tecan, Switzerland). Each culture was tested in triplicates and repeated at least twice independently.

## Protein affinity pull-down assay

His-tagged protein affinity pull-down assay was performed using the recombinant TenA with N-terminal 6 × His tag [72]. The *tenA* gene was cloned into plasmid pET28a to produce recombinant TenA with a N-terminal 6 × His-tag. The coding sequence of *tenA* was amplified with primer pair Pr18650 and Pr18651 from ST606, digested with NdeI and XhoI, and cloned into the multiple cloning site (MCS) of pET28a to generate pTH16586 (pET28a-*tenA*). The construct was then transformed in *E. coli* BL21-DE3 for the inducible expression of TenA with IPTG. Recombinant TenA was purified using Ni-Sepharose resin (GE Healthcare, USA) according to the supplier's instructions. *E. coli* BL21-DE3 with pET28a was cultured in 1 L of LB medium at 37°C, 180 rpm to an $OD_{600 \text{ nm}}$ of 0.5. Bacterial culture was induced for protein expression by the addition of 1 mM IPTG and incubated at 16°C, 180 rpm overnight. Cells were then harvested, washed twice with pre-chilled PBS, and resuspended in 50 ml of binding buffer (10 mM Tris-HCl, pH 8.0, 300 mM NaCl supplemented with 25 mM imidazole, 5 μg/ml DNase I, 10 μg/ml RNase A and one tablet of EDTA-free protease inhibitor (Roche, Sweden)). The cell suspension was lysed using an EmulesiFlex (Jnbio, China) at $1.4 \times 10^3$ Pa. The soluble fraction containing TenA was subsequently purified using Ni-Sepharose resin and eluted with elution buffer (50 mM Tris-Cl pH 8.0, 300 mM NaCl, 250 mM imidazole). Protein concentration was determined with the BCA assay kit (Solarbio, China).

Ni-Sepharose resin (GE Healthcare, USA) was used to immobilize the His-tagged TenA to capture the potential proteins (prey) that interact with TenA. Specifically, 300 μg of the recombinant TenA was mixed with 200 μl Ni-Sepharose resin in 400 μl buffer A (20 mM HEPES (PH 8.0), 100 mM NaCl) containing one tablet of EDTA-free protease inhibitor, and incubated at 4°C for 1 hour. For cell lysate preparation, pneumococcal strain ST556 was cultivated in 1 L of THY medium to an $OD_{620 \text{ nm}}$ of 0.5. Bacterial cultures were collected and washed twice with pre-chilled buffer A. The pellet was resuspended by 20 ml of pre-chilled buffer A, disrupted using an EmulesiFlex at $1.4 \times 10^3$ Pa and then centrifuged to discharge the cell debris. 30 mg of the supernatant was added into the TenA-Ni-Sepharose resin samples and tumbled end over end at 4°C for three hours. Subsequently, the samples were washed with 15 ml washing buffer (20 mM HEPES (PH8.0), 100 mM NaCl, 10 mM $MgCl_2$), and eluted by incubating with 200 μl elution buffer (20 mM HEPES (PH8.0), 100 mM NaCl, 10 mM $MgCl_2$, 250 mM imidazole) for 30 minutes at 4°C. The eluted fraction was assessed by liquid chromatography-tandem mass spectrometry analysis (LC-MS/MS) using a Thermo-Dionex Ultimate 3000 HPLC system combined with the Thermo Orbitrap Fusion mass spectrometer. The spectra from each liquid chromatography–tandem mass spectrometry run were searched against *S. pneumoniae* ST556 database using Proteome Discovery searching algorithm (v1.4) [73]. The same reaction system without bait was used as a negative control. Each experiment was repeated independently at least twice.

## Bacterial adenylate cyclase-based two-hybrid (BACTH) assay

Interactions between TenA and its potential interaction targets were determined by bacterial adenylate cyclase-based two-hybrid (BACTH) system as described [72]. The basic methodology to characterize interactions between two proteins with the BACTH technique involves three main steps: firstly, the proteins of interest are genetically fused to two fragments, T25 and T18, of the catalytic domain of the adenylate cyclase toxin from *Bordetella pertussis*. The *tenA* gene was amplified by PCR using appropriate primers and cloned into pUT18C plasmid to generate T18-TenA; The potential targets were individually sub-cloned into pKT25 vector plasmid to generate T25-X. Secondly, two different recombinant plasmids encoding the T25-X and T18-TenA hybrid proteins were co-transformed into competent *E. coli* Δ*cya* reporter

strain BTH101. If the T25 and T18 fragments are fused to proteins that interact, adenylate cyclase activity will be reconstituted, resulting in production of cAMP to regulate *mal* regulon. Thus, the colony phenotypes of transformants could be plated on MacConkey/maltose agar containing 100 μg/ml ampicillin, 100 μg/ml kanamycin and 0.5 mM IPTG for at least 72 hours. Thirdly, the affinity between two hybrid proteins was quantified by measuring the β-galactosidase activity. In brief, eight independent colonies from each set of transformants were cultivated in 2 ml LB broth supplemented with appropriate antibiotics. After 16 hours, the overnight cultures were diluted 5-fold using M63 medium. The diluted cultures were transferred into a 96-well plate to record the $OD_{595\ nm}$ absorbance with a microplate reader. Meanwhile, the same volume diluted cultures were transferred to a new 96-well plate to permeabilize by adding 0.2% SDS and 0.5% chloroform. After 30 min permeabilization at room temperature, the permeabilized cells are added into PM2 buffer (70 mM $Na_2HPO_4$, 30 mM $NaH_2PO_4$, 1 mM $MgSO_4$, 0.2 mM $MnSO_4$ and 100 mM β-mercaptoethanol) containing 0.1% o-nitrophenol-β-galactoside (ONPG) to start the enzymatic reactions. The reaction was stopped by adding 50 μl of 1 M $Na_2CO_3$ and the $OD_{405\ nm}$ absorbance data were recorded to calculate the relative units (RUs) of β-galactosidase activity.

### Protein structure prediction by AlphaFold3

Predicted protein complexes were modeled based on the amino acid sequences using the AlphaFold3 server (https://www.alphafoldserver.com) [36]. Initial models of TenA, SpxA1 and RimM were first built using the AlphaFold3 server. Prediction of each complex was generated. The predicted complexes were further analyzed and visualized in ChimeraX with structural refinement performed on the top-ranked prediction based on model confidence score [74].

### Statistical analysis

Exact values of significance are indicated in all figures. All analyses were performed by Graph-Pad Prism 9.0 (CA, USA). All experiments were performed at least twice, except for SMRT sequencing. The colony ratio data were statistically analyzed by two-sided Chi-square test (means); qRT-PCR, $hsdS_{A1}$ mRNA quantification and *hsdS* allelic configuration data by two-tailed unpaired parametric *t* test; protein enrichment analysis using Fisher's exact test. The relevant data were presented as the mean ± s.d. of three replicates in a representative experiment. Significant differences were defined by *P* values of $< 0.05$ (∗), $< 0.01$ (∗∗), $< 0.001$(∗∗∗) and $< 0.0001$ (∗∗∗∗).

### Supporting information

**S1 Fig. Colony phases of *tenA*-overexpressed variants in P384 and ST877 strain. (A)** Relative transcriptional expression of *tenA* in *tenA*-overexpressed derivatives of strains P384 (serotype 6A) and ST877 (serotype 35B) were detected and presented as in Fig 1D. **(B)** Colony phenotypes and ratio between O and T colonies in the *tenA*-overexpressed derivatives of strains P384 and ST877.
(TIF)

**S2 Fig. Schematic illustration of detection of the *hsdS* allelic configuration.** The genes encoding the restriction enzyme (*hsdR*), DNA methyltransferase (*hsdM*), sequence recognition proteins (*hsdS*<sub>A</sub>, *hsdS*<sub>B</sub> and *hsdS*<sub>C</sub>) and invertase (*psrA*) are depicted at the top. The promoter and rho-independent transcription terminator are indicated by an arrow and a hairpin. The allelic variants of the *hsdS*<sub>A</sub> gene are depicted below. Three pairs of inverted repeats are indicated by colored arrows (IR1: yellow, IR2: green and IR3: white). PCR primers for detecting

total mRNA of *hsdS* (P1, P2) and *hsdS*$_{A1}$ (P3, P4) are indicated.
(TIF)

**S3 Fig. Similarity between SpxA1-TenA system with GinC system in *S. pyogenes*. (A)** Alignment between SpxA1-TenA system in *S. pneumoniae* and SPY_RS05235-RS05235 system in *S. pyogenes*. TenA in *S. pneumoniae*, SpxA1 in *S. pneumoniae*, GinC (*SPY_RS05230*) and Anti-GinC (*SPY_RS05235*) in *S. pyogenes* M1 (NC_002737.2) were aligned by Smith-Waterman method. Identical or similar amino acids are indicated by blue box. The black color indicates the amino acids that are not similar or the gaps. **(B)** Alignment between SpxA1 and SpxA2 in *S. pneumoniae*. SpxA1 and SpxA2 were aligned and presented as in (A).
(TIF)

**S4 Fig. Role of the Spx proteins in regulation of *hsdS* inversions. (A)** The relative mRNA levels of *spxA1* in the *spxA1* deletion and overexpression variants were detected by RT-qPCR as in Fig 1C. The mRNA levels were presented as relative values to that of WT (left panel) and JC1 control strain (right panel), respectively. **(B)** The relative mRNA levels of *spxA1-tenA* in *spxA2* mutant and revertant were detected by RT-qPCR as in Fig 1C. **(C)** Relative abundance of the *hsdS*$_{A1}$ mRNA in *spxA1* and *spxA2* mutants. **(D)** Detection of TenA interactions with SpxA1 by bacterial two-hybrid (BACTH) assay. Colonies on the MacConkey/maltose plates (left panel) and β-galactosidase activity (right panel) are shown for each reporter strain. PC, positive control (pKT25-*zip* and pUT18C-*zip*). NC, negative control (empty vectors pKT25 and pUT18C). The mean ± s.d. of three values (from three individual experiments) of each strain is presented in a single bar. **(E)** Predicted interaction of the SpxA1-TenA complex. The structural models and surface electro statics were Predicted, presented and labelled as in Fig 6C. TenA and SpxA1 are Indicated as blue and pink.
(TIF)

**S5 Fig. Molecular functions of TenA-associated proteins.** Each slice lists the numbers of proteins functions based on the assignment to molecular function categories in the Gene Ontology (GO, available at www.geneontology.org/).
(TIF)

**S6 Fig. Regulatory network of *hsdS* inversion by ComX. (A**) Colony phenotype and ratio between O and T colonies of the TenA-targeted protein mutants. **(B)** Verification of the expression of the *psrA*, *rib*, *rafG*, and *arcA* locus in the *tenA*-overexpressed variant and *comX* deletion mutants. **(C)** Relative expression of *psrA* in the *psrA*-overexpressed variants. **(D** to **F**) Relative abundance of the *hsdS*$_{A1}$ mRNAs (**D**), ratio between O and T colonies (**E**) and *hsdS*$_A$ allelic configurations (**F**) of the *rib* locus mutants.
(TIF)

**S7 Fig. Prediction of three-dimensional structure of TenA and RimM. (A and B)** Cartoon model of the structure of TenA (**A**) and RimM (**B**) predicted by AlphaFold3. Red (low confidence) and blue (high confidence) colors show the pLDDT values per position. The N-terminal domain (N-ter) and C-terminal domain (C-ter) of TenA and RimM are indicated. The pLDDT values are marked.
(TIF)

**S1 Table. The opacity ratio of ST556 derivatives.**
(DOCX)

**S2 Table. Methylation sequences specified by the Spn556I MTase.**
(DOCX)

**S3 Table. Methylation sequences specified by the Spn556III MTase.**
(DOCX)

**S4 Table. The relative abundance of the TenA targets.**
(XLSX)

**S5 Table. The transcripts of the *tenA* and *comX1X2* mutants.**
(XLSX)

**S6 Table. Basic info of AT-rich region containing strains.**
(XLSX)

**S7 Table. Bacterial strains or plasmids used in this study.**
(DOCX)

**S8 Table. List of primers used in this study.**
(DOCX)

**S9 Table. Information for constructions of strains in this study.**
(DOCX)

## Acknowledgments

We thank Tsinghua research platforms for assistance in protein mass spectrometry (Center for Proteomics) and qRT-PCR (Medical Microbiology Core Facility); Xiaolin Tian and Dingfei Yan for the analysis of mass spectrometry data; Jiao Hu for the analysis of RNA-sequencing data.

## Author Contributions

**Conceptualization:** Shaomeng Wang, Juanjuan Wang, Jing-Ren Zhang.

**Data curation:** Shaomeng Wang, Xiu-Yuan Li.

**Formal analysis:** Shaomeng Wang.

**Funding acquisition:** Juanjuan Wang, Jing-Ren Zhang.

**Investigation:** Shaomeng Wang, Xiu-Yuan Li, Mengran Zhu, Juanjuan Wang, Jing-Ren Zhang.

**Methodology:** Shaomeng Wang, Xiu-Yuan Li, Haiteng Deng, Juanjuan Wang, Jing-Ren Zhang.

**Project administration:** Jing-Ren Zhang.

**Software:** Shaomeng Wang.

**Supervision:** Juanjuan Wang, Jing-Ren Zhang.

**Validation:** Juanjuan Wang, Jing-Ren Zhang.

**Visualization:** Shaomeng Wang, Jing-Ren Zhang.

**Writing – original draft:** Shaomeng Wang.

**Writing – review & editing:** Shaomeng Wang, Juanjuan Wang, Jing-Ren Zhang.

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
