## [Decision Letter · Decision Letter 0]

28 Oct 2024

PPATHOGENS-D-24-01936The SpxA1-TenA Toxin-Antitoxin System Regulates Epigenetic Variations of Streptococcus pneumoniae by Targeting Protein SynthesisPLOS Pathogens Dear Jing-Ren, Thank you for submitting your manuscript to PLOS Pathogens. After careful consideration, we feel that it has merit but does not fully meet PLOS Pathogens's publication criteria as it currently stands. Therefore, we invite you to submit a revised version of the manuscript that addresses the points raised during the review process. In addition to the comments of reviewers, please consider addressing the following I (JL) noted: In Fig. S4A and B. the ∆spxA2 exhibits increased spxA1 transcription and also a modest but significant decrease in hsdS_A1_ mRNA. In my reading, these findings are not reflected in the text, lines 236-239. In addition, I believe that there should be “∆” placed before "spxA1^rev^" and spxA2^rev^" in Fig. S4A. (Note also that I believe that Fig. S4 indicates that you already constructed a spxA1-complemented strain that Reviewer 3 suggested to include in order to reinforce the conclusions of Fig. 4B.) Please submit your revised manuscript within 30 days Dec 27 2024 11:59PM. If you will need more time than this to complete your revisions, please reply to this message or contact the journal office at plospathogens@plos.org. Please include the following items when submitting your revised manuscript:*
A rebuttal letter that responds to each point raised by the editor and reviewer(s). You should upload this letter as a separate file labeled 'Response to Reviewers'. This file does not need to include responses to any formatting updates and technical items listed in the 'Journal Requirements' section below.*
A marked-up copy of your manuscript that highlights changes made to the original version. You should upload this as a separate file labeled 'Revised Manuscript with Track Changes'.*
An unmarked version of your revised paper without tracked changes. You should upload this as a separate file labeled 'Manuscript'. If you would like to make changes to your financial disclosure, competing interests statement, or data availability statement, please make these updates within the submission form at the time of resubmission. Guidelines for resubmitting your figure files are available below the reviewer comments at the end of this letter. We look forward to receiving your revised manuscript. Kind regards, John M LeongAcademic EditorPLOS Pathogens Alice PrinceSection EditorPLOS Pathogens Michael Malim

Editor-in-Chief

PLOS Pathogens

orcid.org/0000-0002-7699-2064 **Journal Requirements:** **Additional Editor Comments (if provided):****Reviewers' Comments:** Reviewer's Responses to Questions

**Part I - Summary**

Reviewer #1: The present study demonstrates that epigenetic traits of the pathogen Streptococcus pnumoniae are controlled by a novel Toxin-Antitoxin system. It is a new and important study in the complex biology of this bacterium, and represents an important step forward to our understanding of bacterial virulence.

Reviewer #2: This a highly significant and novel study with many strengths and only minor grammatical weaknesses. The study is really a tour de force of investigation into how phase variation in pneumococcus is regulated. It could easily have been broken up into two or three smaller papers, but instead, the authors present it as one coherent, rigorous study.

Reviewer #3: Wang and Li et al identified a new T/A system, consisting of SpxAI and TenA, in Streptococcus pneumoniae. Specifically, they found that TenA represses RimM maturation and ComX levels. Additionally, this system controls phases of the epigenome. The findings are rigorous, novel, and explained extremely well which is critical for a journal with a broad audience.

**Part II – Major Issues: Key Experiments Required for Acceptance**

Reviewer #1: The mansuscript is well presented and, in my opinion, does not need further experiments.

Reviewer #2: None

Reviewer #3: 1. Major comment: In lines 256-258 the authors state their findings confirm that TenA is transcriptionally repressed by SpxA1. How do the authors think this is happening? Is this just because the proteins are on the same transcript? To explore this finding, it would be useful for the authors to show a knock in of spxAI in a neutral locus represses tenA transcription levels (see Fig4B). Regardless of this experiment, Fig4B needs a spXAI complemented control.

**Part III – Minor Issues: Editorial and Data Presentation Modifications**

Reviewer #1: The paper is very well rittent, the Figures are very clear.

Reviewer #2: 1. Line 469: The journal may not allow “Data not shown”. If you have the data, it should go in supplement. Alternatively, delete this sentence.

2. Line 490: With respect to the sentence “In particular, the vast majority of the protein hits in the TenA affinity pull-down were associated with protein synthesis.”, is it possible that TenA actually only binds RimM, but RimM is bound to these other ribosomal proteins?

3. Table 1. Is a p value of zero a legitimate value to report? I believe it is better to report p < [smallest representable value].

4. Delete “interestingly” here and elsewhere (line 295). That is opinion and is not very scientific.

5. Line 121: Change “insertion sequence” to “inserted sequence”. An insertion sequence is technically a transposable element in bacteria.

6. Line 148: Delete “finally”.

7. Lines 158, 162, 341, 452: Why is “via” italicized? It shouldn’t be.

8. Line 239: I believe you want to delete “as SpxA1” at the end of the sentence. Otherwise, rewrite the sentence to be more precise.

9. Line 256: “have confirmed” is a little too strong. I suggest changing to “suggest” or “indicate”.

10. Line 264: Change “trial” to “experiment”.

11. Line 304: Start sentence with “This”.

12. Line 319: Change “mostly” to “most”.

13. Line 346: Change “persisted robustly” to “abundant”, or something more clear.

14. Line 360: Change “TIRGR4” to “TIGR4”.

15. Line 378: Strain not “stain”.

16. In the RNA isolation and sequencing section in Materials and Methods, we rRNA removed? This wasn’t mentioned. If not, then fine.

17. Line 657: Delete “and vitality”.

Reviewer #3: Minor comments:

a. Fig1A: Is the arrow in the bottom right panel supposed to be blue? The colony looks opaque.

b. What does the red arrow mean in Fig2B.

c. Fig5E lacks a Y axis label

PLOS authors have the option to publish the peer review history of their article (what does this mean?). If published, this will include your full peer review and any attached files.

Reviewer #1: **Yes: **Manuel Espinosa

Reviewer #2: No

Reviewer #3: No

---

## [Editor Report · Decision Letter 1]

2 Dec 2024

Dear Jing-Ren,

We are pleased to inform you that your manuscript 'The SpxA1-TenA Toxin-Antitoxin System Regulates Epigenetic Variations of *Streptococcus pneumoniae* by Targeting Protein Synthesis' has been provisionally accepted for publication in PLOS Pathogens.

Best regards,

John M Leong

Academic Editor

PLOS Pathogens

Alice Prince

Section Editor

PLOS Pathogens

Michael Malim

Editor-in-Chief

PLOS Pathogens

orcid.org/0000-0002-7699-2064
---

## [Editor Report · Acceptance letter]

17 Dec 2024

Dear Dr. Zhang,

We are delighted to inform you that your manuscript, "The SpxA1-TenA Toxin-Antitoxin System Regulates Epigenetic Variations of *Streptococcus pneumoniae* by Targeting Protein Synthesis," has been formally accepted for publication in PLOS Pathogens.

Best regards,

Sumita Bhaduri-McIntosh

Editor-in-Chief

PLOS Pathogens

orcid.org/0000-0003-2946-9497

Michael Malim

Editor-in-Chief

PLOS Pathogens

orcid.org/0000-0002-7699-2064